

# Double soft theorem for generalised biadjoint scalar amplitudes

**Md. Abhishek[⋆], Subramanya Hegde, Dileep P. Jatkar and Arnab Priya Saha**

Harish-Chandra Research Institute, Homi Bhabha National Institute (HBNI),
Chhatnag Road, Jhunsi, Allahabad, India 211019

⋆ mdabhishek@hri.res.in

## Abstract

We study double soft theorem for the generalised biadjoint scalar field theory whose amplitudes are computed in terms of punctures on $\mathbb{CP}^{k-1}$. We find that whenever the double soft limit does not decouple into a product of single soft factors, the leading contributions to the double soft theorems come from the degenerate solutions, otherwise the non-degenerate solutions dominate. Our analysis uses the regular solutions to the scattering equations. Most of the results are presented for $k = 3$ but we show how they generalise to arbitrary $k$. We have explicit analytic results, for any $k$, in the case when soft external states are adjacent.



# 1   Introduction and Summary

Our understanding of the scattering amplitudes has improved manifolds in the last couple of decades. That the formulae for amplitudes simplify significantly if we use the spinor helicity formalism dates back to the mid-eighties [1–3]. However, recent interest in this direction came from the twistor formulation of the $\mathcal{N} = 4$ super-Yang-Mills theory [4, 5], leading to the BCFW formulation of the scattering amplitudes [6–8]. The representation of the $\mathcal{N} = 4$ super-Yang-Mills theory in terms of Grassmannians [9–11] as well as polytope realisation of the scattering amplitudes [9,12–24], and the Cachazo-He-Yuan(CHY) formulation of the scattering amplitudes [25–30] gave further impetus to unraveling their underlying structure.

The space of Mandelstam invariants of $n$ massless particles is isomorphic to the moduli space of $\mathbb{CP}^1$ with $n$ marked points, the CHY formulation therefore naturally involves the punctured Riemann surfaces and in particular, for tree level $n$-point amplitudes they were written in terms of the integral over the moduli space of $n$-punctured sphere $\mathbb{CP}^1$. The CHY amplitudes were generalised in different ways, which led to developments such as the ambitwistor string theory [31–39], and the positive Grassmannian formulation [9, 10, 40–42] of string theory and field theory amplitudes. Recently one of the promising generalisations involved replacing $\mathbb{CP}^1$ by $\mathbb{CP}^{k-1}$ [43,44]. This generalisation, in some sense, is straightforward from the Grassmannian point of view but is not at all obvious from the field theory point of view. In other words, going from $Gr(2, n)$ to $Gr(k, n)$ seems like a natural thing to attempt from the Grassmannian picture, but it is not clear what kind of field theories for which this generalised formulation of scattering amplitudes applies. In fact, it was known that $N^{k-2}$MHV amplitudes can be written in terms of $Gr(k, n)$ [45] well before this generalization [43, 44] was proposed, which indicates that this formulation could be useful in unravelling the structure of loop amplitudes. Nevertheless, the generalization of the biadjoint scalar theory with $\mathbb{CP}^{k-1}$ kinematic space begs for a field theory formulation, which presumably would give a better insight and provide more physical methods for dealing with these amplitudes. There has been some progress in understanding these amplitudes for $k > 2$ in terms of planar arrays of Feynman diagrams [46–48].

In this paper, we will study double soft limits of amplitudes in the generalised biadjoint scalar field theory. In [49] it was shown that single soft limits of the $(k, n)$ amplitudes admit identification with the $(2, k + 2)$ amplitudes. Whether such an identification can be generalised to multiple soft limits is an interesting question to explore. Here we will take the first step in this direction by computing double soft factors in the generalised biadjoint scalar theories. Since $(k, k + 1)$ amplitudes are trivial by a gauge choice, the first non-trivial single soft limit is applicable to $(k, k + 2)$ amplitudes. Naturally, the first non-trivial case of soft limit with $m$ number of soft external legs can be applied to $(k, k + m + 1)$ amplitudes. Using the Grassmannian duality these amplitudes are related to $(m + 1, k + m + 1)$ amplitudes. In particular, for the double soft ($m = 2$) factors, we naturally expect them to be related to $(3, k + 3)$ amplitudes. In addition note that as the dimension of the moduli space for $(k, n)$ amplitudes is $(k - 1)(n - k - 1)$, the multi soft limits mentioned above probe the maximal codimension boundaries of the $m(k - 1)$ dimensional moduli space.

With the motivation given above, we will explore the structure of double soft factors in arbitrary $(k, n)$ case. We will, however, give a more comprehensive account of our results for $k = 3$ case, *i.e.*, when the amplitudes are described as punctures on $\mathbb{CP}^2$, and then generalise them to arbitrary $k$. We will, however, not explicitly use the positive Grassmannian formulation here. Single and multiple soft theorems in a variety of theories including gluons and gravitons have been worked out using the CHY formalism [50–58]. The biadjoint scalar field theory has been the main arena for exploration, both in the CHY formalism as well as its generalisation to higher $k$ [43,44]. By virtue of being a biadjoint field, its amplitudes are parametrised in terms of two sets of adjoint indices, we will denote them as $\alpha$ and $\beta$, respectively. The amplitudes can have independent color ordering with respect to these two adjoint indices. Throughout this work, we will choose both $\alpha$ and $\beta$ to be the canonical ordering, that denoted as I, we will use this notation for the sake of brevity.

The solutions to scattering equations for $k \geq 3$ can be categorised into two types, regular and singular solutions, based on the behaviour of kinematic determinants in the soft limit. We will exclusively be focusing on the regular solutions. In addition, in this work we will be interested in the leading contribution to the soft theorem and hence we will pay attention to those configurations that give dominant contribution to the single or double soft limit. We find that, in the double soft limit, those configurations that do not factorise into a product of two single soft configurations have dominant contributions coming from the degenerate solutions. In all these cases, the non-degenerate solutions lead to subleading contributions and hence, in this paper, we do not take these cases into account. The non-decoupling double soft configurations for arbitrary $k$ occur when two soft particles in the amplitude are such that separation between them, for the canonical ordering, is not more than $k - 2$. For example, for $k = 2$, they must be adjacent external states, whereas for $k = 3$ they can be adjacent or the next to adjacent. For any configuration with index separation larger than this leads to the double soft factor which is a product of two single soft factors.

The double soft limit contains two main cases, simultaneous double soft limit and consecutive double soft limit. In the simultaneous double soft case, two external states are taken soft at an equal rate, and in the latter case, one state becomes soft at a faster rate than the other. The leading contribution to the simultaneous double soft limit comes from the degenerate solutions which have singularities corresponding to collision of two soft punctures or corresponding to collinear limit of two soft punctures with a hard puncture. In the latter case, two soft states scale differently with either $\tau_1 \ll \tau_2$ or $\tau_1 \gg \tau_2$, where soft limit corresponds to $\tau_i \to 0$ in a sequential manner. We also establish that simultaneous double soft limit can be arrived at by taking $\tau_1 = \tau_2$ limit of the consecutive limit. In some sense, this is a consistency check for our computation of the double soft limits.

For arbitrary $k$, the simultaneous double soft factor for the adjacent soft external states is

given by,

$$\mathsf{S}_{\mathrm{DS}}^{(k)} = \frac{1}{\displaystyle\sum_{1 \le a_1 \cdots < a_{k-2} \le n-2} s_{a_1 \cdots a_{k-2} \, n-1 \, n}} \mathsf{S}^{(k-1)}\left(s_{a_1 \cdots a_{k-2} \, m} \to s_{a_1 \cdots a_{k-2} \, n-1 \, n}\right) \mathsf{S}^{(k)} \,, \qquad (1.1)$$

where $s_{a_1 \cdots a_k}$ are generalised Mandelstam variables. In Eq. (1.1), we have taken the external states $n$ and $n-1$ soft, and the argument of the single soft factor $\mathsf{S}^{(k-1)}$ signifies that the soft label $m$ for $\mathsf{S}^{(k-1)}$ is replaced by a composite label '$n\ n-1$'. The single soft factor, $\mathsf{S}^{(k)}$ is defined with a shifted propagator $s_{a_1 \cdots a_{k-1} \, n-1} + s_{a_1 \cdots a_{k-1} \, n}$. This is the main result of this paper. We also find, for any $k$, that the leading contribution to the double soft factor scales as $\tau^{-3(k-1)}$ in the $\tau \to 0$ limit.

We generalise our analysis to the next to adjacent soft external states for $k = 3$, where we encounter a high degree polynomial equation to solve for the punctures in terms of the generalised Mandelstam variables. We, however, do not have generic explicit solutions to this polynomial equation. In the case of $k = 3$, we show that the double soft factor for the next to next to adjacent soft external states factorises into a product of two $k = 3$ single soft factors for each of the soft external states.

The paper is organised as follows: Section 2 is more of a review of the $k = 2$ CHY formalism. In this section we will study single and double soft limits of $n$ point amplitudes in the biadjoint scalar field theory. We, however, will present the results for the double soft limit of the amplitudes with non-adjacent soft punctures. As in the literature, for an arbitrary $k$, we will use phrases like punctures on $\mathbb{CP}^{k-1}$, external states, and external particles interchangeably. In section 3, we will discuss the single soft theorem for arbitrary $k$. After setting up the notation, we will consider single soft theorem in $k = 3$ case and analyse collision as well as collinear singularities. We then generalise these results to arbitrary $k$. This section summarises the results of [49], but the method spelt out in this section is useful for generalisation to the double soft limit. Section 4 contains a detailed study of the double soft theorem for $k = 3$, where we consider two external soft states to be adjacent. Section 5 contains generalisation of the double soft theorem for adjacent external states to arbitrary $k$. In section 6, we revert to the $k = 3$ case and study the next to adjacent double soft limit of $n$ point amplitudes and in 7, we study the next to next to adjacent double soft limit. We conclude with section 8, which contains a discussion on applications and possible extensions of these results.

## 2 Soft Theorems for Biadjoint Scalar Field for $k = 2$

In this section, we review the single and double soft limits of biadjoint scalar amplitudes for $k = 2$ in the CHY formalism. As mentioned in the introduction, we will take both $\alpha$ and $\beta$ to be canonically ordered, $\mathbb{I} = 1, 2, \ldots n$. Here we shall consider soft limits in $m_n^{(2)}(\mathbb{I}|\mathbb{I})$. In the resulting lower point amplitude canonical ordering would mean labels are arranged in ascending order of magnitude after omitting the soft particles.

### 2.1 Single soft limit

We will begin with the familiar single soft limit of the $n$-point amplitude of the biadjoint scalar field theory. We will present the computations in both homogeneous and inhomogeneous coordinates, which helps us set up the notation for the rest of the paper. In the homogeneous coordinates, we can express the punctures on $\mathbb{CP}^1$ by,

$$\sigma_a = \begin{pmatrix} Z_a^1 \\ Z_a^2 \end{pmatrix} = Z_a^1 \begin{pmatrix} 1 \\ x_a \end{pmatrix}, \qquad x_a = \frac{Z_a^2}{Z_a^1} \,, \qquad (2.1)$$

where $Z_a$ are homogeneous coordinates and $x_a$ are projective coordinates defined in the coordinate patch where $Z_a^1$ is non-vanishing. It is convenient to introduce a potential function [59],

$$\mathcal{S}^{(2)} = \sum_{b \neq a} s_{a\,b} \log |a\ b| , \qquad |a\ b| = \begin{vmatrix} 1 & 1 \\ x_a & x_b \end{vmatrix} , \tag{2.2}$$

whose extremisation gives the scattering equations, which in the projective coordinates take the form,

$$E_a := \sum_{b \neq a} \frac{s_{a\,b}}{x_a - x_b} = \frac{\partial}{\partial x_a} \sum_{b \neq a} s_{a\,b} \log |a\ b| = 0 , \tag{2.3}$$

where $s_{a\,b}$ are the Mandelstam variables. At this stage although we can treat the Mandelstam variables in terms of specific functions of momenta, e.g., $s_{a\,b} = 2k_a \cdot k_b$, for the purpose of generalisation to arbitrary $k$ we will keep them generic without referring to explicit dependence on momenta. We now define,

$$\begin{aligned}
E_a' &:= \sum_{b \neq a} \frac{s_{a\,b}}{|a\ b|} |X\ b| \\
&= \sum_{b \neq a} \frac{s_{a\,b}}{|a\ b|} (x_b - x_a + x_a - x) \\
&= |X\ a| \sum_{b \neq a} \frac{s_{a\,b}}{|a\ b|} \\
&= - |X\ a| E_a .
\end{aligned} \tag{2.4}$$

Here $X$ is an arbitrary reference vector on $\mathbb{CP}^1$ and we will denote it by $X = \begin{pmatrix} 1 \\ x \end{pmatrix}$. In the third equality in Eq. (2.4) we have used the condition of momentum conservation. It follows from the Eq. (2.4) that the delta function for $a$-th scattering equation can be expressed as,

$$\delta(E_a) = - |X\ a|\, \delta(E_a'). \tag{2.5}$$

In order to take the single soft limit, we need to choose soft momenta for one of the external legs. Without loss of generality, we will choose $n$-th particle momentum to be soft. Since the external momenta are in one to one correspondence with the punctures on $\mathbb{CP}^1$, we will use the words momentum and punctures interchangeably. In the later sections, where there is no clear description in terms of momenta, we will only use the term punctures. We will denote the soft puncture by $\sigma$ in homogeneous coordinates. In this coordinates the one-form on $\mathbb{CP}^1$ can be written as,

$$\begin{aligned}
(\sigma\, d\sigma) &= \begin{vmatrix} Z^1 & dZ^1 \\ Z^2 & dZ^2 \end{vmatrix} \\
&= (Z^1)^2 dx_n, \qquad x_n = \frac{Z^2}{Z^1} .
\end{aligned} \tag{2.6}$$

Since the $n$-th external state has soft momentum, the $n$-particle amplitude factorises into $n-1$-particle amplitude times the soft factor. We therefore denote, $m_n^{(2)}(\mathbb{I}|\mathbb{I}) = \mathbb{S}^{(2)}\, m_{n-1}(\mathbb{I}|\mathbb{I})$, where the soft factor given by,

$$\mathbb{S}^{(2)} = - \oint \frac{(\sigma\, d\sigma)(X\, \sigma)}{\sum_{b \neq n} \frac{s_{n\,b}(X\,b)}{(\sigma\,b)}} \left[ \frac{(n-1\,1)}{(n-1\,\sigma)(\sigma\,1)} \right]^2$$

$$= -\oint \frac{dx_n\,(x_n - x)}{\sum\limits_{b \neq n} \frac{s_{nb}(x_b - x)}{x_b - x_n}} \left[ \frac{x_1 - x_{n-1}}{(x_n - x_{n-1})(x_1 - x_n)} \right]^2$$

$$= \frac{1}{s_{n\,n-1}} + \frac{1}{s_{n\,1}}\,. \tag{2.7}$$

In the last line we have used the residue theorem to evaluate contributions coming from simple poles at $x_n = x_{n-1}$ and $x_n = x_1$.

## 2.2  Double soft limits

We will now study simultaneous soft limits with two external soft momenta. For the biadjoint scalars there are qualitatively two different ways of taking simultaneous soft limits: in a given color ordered arrangement, either two adjacent states are taken soft, or two non-adjacent states are taken soft. As we will see below in the former case, the soft factor scales as $\tau^{-3}$, whereas in the latter case, the scaling is $\tau^{-2}$.

### 2.2.1  Adjacent soft limit

As in the previous subsection, we will continue to take $k_n$, the momentum of $n$-th particle soft. In addition we will consider $k_{n-1}$, the momentum of $(n-1)$-th particle, to be soft as well. We implement these soft limits by scaling the soft momenta as $k_n = \tau \hat{k}_n$ and $k_{n-1} = \tau \hat{k}_{n-1}$ and take $\tau \to 0$. As a result the Mandelstam variables scale in the following manner, $s_{n-1\,a} = \tau \hat{s}_{n-1\,a}$ and $s_{n\,a} = \tau \hat{s}_{n\,a}$, on the other hand $s_{n-1\,n} = \tau^2 \hat{s}_{n-1\,n}$. Using this scaling property we decompose the scattering equations based on their scaling property in soft limit as,

$$E_a = \sum_{\substack{b=1 \\ b \neq a}}^{n-2} \frac{s_{a\,b}}{x_a - x_b} = 0\,, \qquad a \in \{1, 2, \dots n-2\}$$

$$E_{n-1} = \sum_{b=1}^{n-2} \frac{s_{n-1\,b}}{x_{n-1} - x_b} + \frac{s_{n-1\,n}}{x_{n-1} - x_n} = 0\,,$$

$$E_n = \sum_{b=1}^{n-2} \frac{s_{n\,b}}{x_n - x_b} - \frac{s_{n-1\,n}}{x_{n-1} - x_n} = 0\,. \tag{2.8}$$

The integral representation of the soft factor is,

$$S_{DS}^{(2)} = \int dx_n\,\delta(E_n) \int dx_{n-1}\,\delta(E_{n-1}) \left[ \frac{x_{n-1} - x_1}{(x_{n-2} - x_{n-1})(x_{n-1} - x_n)(x_n - x_1)} \right]^2\,. \tag{2.9}$$

The solutions to Eq.(2.8) fall into two categories [54], the non-degenerate solutions where $|x_{n-1} - x_n| \sim \mathcal{O}(\tau^0)$ and the degenerate solutions where $|x_{n-1} - x_n| \sim \mathcal{O}(\tau)$. In the adjacent soft limit, contribution from the degenerate solutions dominate over those of the non-degenerate ones. Since we are interested in picking up the leading contribution, we will consider only the degenerate solutions.

In the degenerate case we make a change of variables,

$$x_{n-1} = \rho + \xi\,, \qquad x_n = \rho - \xi\,, \tag{2.10}$$

where, $\xi$ is $\mathcal{O}(\tau)$. With this change of variables it is convenient to re-express the delta functions with arguments $E_{n-1}$ and $E_n$ in terms of sum and difference of the two scattering equations. The integral representation of the double soft factor then becomes,

$$S_{DS}^{(2)} = \int d\rho \int d\xi\,\delta(E_{n-1} + E_n)\delta(E_{n-1} - E_n) \left[ \frac{x_{n-2} - x_1}{\xi(x_{n-2} - \rho)(\rho - x_1)} \right]^2\,. \tag{2.11}$$

The $\xi$ integral can be evaluated using the second delta function to localise $\xi$ to its solution [54–58]. This method is not convenient when we study higher $k$ generalisations. Therefore instead of solving for $\xi$, we can use the second delta function to convert $\xi$ integration to a contour integral. We can then use the contour deformation method and pick up poles from the integrand which came out of the Parke-Taylor factor in the soft limit. The pole is seen to be at $\xi = 0$ and in the neighbourhood of this pole we can approximate $E_{n-1} - E_n \approx \frac{s_{n-1\,n}}{\xi}$. The soft factor can now be evaluated as,

$$
\begin{aligned}
S_{DS}^{(2)} &= -\int d\rho \; \delta(E_{n-1} + E_n) \oint_{\substack{\frac{s_{n-1\,n}}{\xi} \\ \{\xi \to 0\}}} \frac{d\xi}{\xi^2} \left[ \frac{x_{n-2} - x_1}{(x_{n-2} - \rho)(\rho - x_1)} \right]^2 \\
&= \frac{1}{s_{n-1\,n}} \oint_{\{\rho \to x_{n-2}, x_1\}} \frac{d\rho}{\sum\limits_{b=1}^{n-2} \frac{s_{n-1\,b} + s_{n\,b}}{\rho - x_b}} \left[ \frac{x_{n-2} - x_1}{(x_{n-2} - \rho)(\rho - x_1)} \right]^2 \\
&= \frac{1}{s_{n-1\,n}} \left[ \frac{1}{s_{n-1\,n-2} + s_{n\,n-2}} + \frac{1}{s_{n-1\,1} + s_{n\,1}} \right].
\end{aligned}
\tag{2.12}
$$

It follows from the last line of Eq. (2.12) that $S_{DS}^{(2)}$ scales as $\tau^{-3}$. This scaling can be also understood from the Feynman diagrams given below, where the blob stands for the $n-3$ point tree diagram.

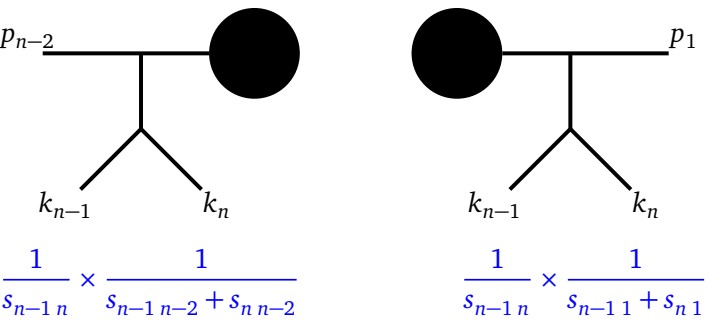

As can be seen from the above diagrams, the intermediate propagator connecting two soft legs with momenta $k_{n-1}$ and $k_n$ with the hard leg having the momentum $p_a$, $a = \{n-2, 1\}$ gives a factor of $\tau^{-2}$ whereas the propagator joining the blob gives a factor of $\tau^{-1}$ making overall scaling of the double soft factor to be $\tau^{-3}$. This scaling may change in theories with the derivative couplings, as factors of soft momenta can also come from the interaction vertices. The power of soft momenta in the denominator is, therefore, reduced and the double soft factors scale as $\tau^{-m}$ where $m \leq 2$ [54–58, 60–66].

### 2.2.2  Non-adjacent soft limit

We will now consider soft limits for two non-adjacent external states. To illustrate this, we consider $n-2$ and $n$ to be soft punctures. Following from our earlier analysis, it is easy to see that $s_{n-2\,a}$ and $s_{n\,a}$ scale as $\tau$, and $s_{n-2\,n} \to \tau^2$ in the limit $\tau \to 0$. The label $a = 1, \cdots, n-3, n-1$ denotes the hard external particles. The soft factor can then be written as,

$$
\begin{aligned}
S_{DS}^{(2)} &= \int dx_n \, \delta(E_n) \int dx_{n-2} \, \delta(E_{n-2}) \left[ \frac{(x_{n-3} - x_{n-1})(x_{n-1} - x_1)}{(x_{n-3} - x_{n-2})(x_{n-2} - x_{n-1})(x_{n-1} - x_n)(x_n - x_1)} \right]^2 \\
&= \left( \frac{1}{s_{n\,n-1}} + \frac{1}{s_{n\,1}} \right) \left( \frac{1}{s_{n-2\,n-3}} + \frac{1}{s_{n-2\,n-1}} \right).
\end{aligned}
\tag{2.13}
$$

It readily follows that $S_{DS}^{(2)}$ in Eq. (2.13) scales as $\tau^{-2}$ and is equal to the product of two single soft factors found in Sec.(2.1). In general if we consider soft limits in any two non-adjacent states, then the double soft factor is the product of individual single soft factors for the corresponding soft external states.

## 3   Single Soft Theorem for Generalised Biadjoint Scalars

In the previous section, we reviewed aspects of $k = 2$ biadjoint scalar theory in the CHY formalism [25–28]. We will now discuss its generalisation where instead of studying punctures on $\mathbb{CP}^1$ as will consider punctures on $\mathbb{CP}^{k-1}$ [44]. From now on, we will refer to external states as punctures on $\mathbb{CP}^{k-1}$ since there is no clear representation of them in terms of momenta.

The scattering equations for any $k$ are obtained by saddle point analysis of the scattering potential function, which is a generalisation of the $k = 2$ scattering potential, given as,

$$\mathcal{S}^{(k)} = \sum_{a_1 < a_2 < \cdots < a_k} s_{a_1 a_2 \cdots a_k} \log(a_1 a_2 \cdots a_k), \tag{3.1}$$

where $(a_1 a_2 \cdots a_k)$ is the determinant,

$$(a_1 a_2 \cdots a_k) = \begin{vmatrix} \sigma_{a_1}^{(1)} & \sigma_{a_2}^{(1)} & \cdots & \sigma_{a_k}^{(1)} \\ \sigma_{a_1}^{(2)} & \sigma_{a_2}^{(2)} & \cdots & \sigma_{a_k}^{(2)} \\ \vdots & \vdots & \ddots & \vdots \\ \sigma_{a_1}^{(k)} & \sigma_{a_2}^{(k)} & \cdots & \sigma_{a_k}^{(k)} \end{vmatrix}, \tag{3.2}$$

and $\{\sigma_a^{(1)}, \sigma_a^{(2)}, \cdots, \sigma_a^{(k)}\}$ are the homogeneous coordinates describing the puncture $\sigma_a$ on $\mathbb{CP}^{k-1}$. The generalised Mandelstam variables, denoted as $s_{a_1 a_2 \cdots a_k}$, are symmetric tensors of rank $k$ and has the property that it vanishes whenever two indices on it are same,

$$s_{a_1 a_2 \cdots a_i \cdots a_i \cdots a_k} = 0, \tag{3.3}$$

and projective invariance of the scattering potential gives,

$$\sum_{a_2 < a_3 < \cdots < a_k \neq a_1} s_{a_1 a_2 \cdots a_k} = 0, \quad \forall a_1. \tag{3.4}$$

These two conditions are generalisations of masslessness condition and momentum conservation of Mandelstam variables to higher $k$.

We will be working in the inhomogeneous coordinates, where the scattering potential function is written as,

$$\mathcal{S}^{(k)} = \sum_{a_1 < a_2 < \cdots < a_k} s_{a_1 a_2 \cdots a_k} \log|a_1 a_2 \cdots a_k|, \tag{3.5}$$

where $|a_1 a_2 \cdots a_k|$ is the determinant,

$$|a_1 a_2 \cdots a_k| = \begin{vmatrix} 1 & 1 & \cdots & 1 \\ x_{a_1}^1 & x_{a_2}^1 & \cdots & x_{a_k}^1 \\ \vdots & \vdots & \ddots & \vdots \\ x_{a_1}^{k-1} & x_{a_2}^{k-1} & \cdots & x_{a_k}^{k-1} \end{vmatrix}, \tag{3.6}$$

and $\{1, x_a^1, x_a^2, \cdots, x_a^k\}$ are the inhomogeneous coordinates for the puncture $\sigma_a$. The scattering equation for arbitrary $k$ can now be given as,

$$
\begin{aligned}
E_a^{(i)} &= \frac{\partial \mathcal{S}^{(k)}}{\partial x_a^i} \\
&= \sum_{a_2 < \cdots < a_k \neq a} \frac{s_{aa_2\cdots a_k}}{|aa_2\cdots a_k|} \frac{\partial}{\partial x_a^i} |aa_2\cdots a_k| = 0, \qquad \forall a = 1, \cdots, n.
\end{aligned}
\tag{3.7}
$$

This equation hold for each $i = 1, \cdots k-1$. Analogous to the $k = 2$ case, the biadjoint scalar amplitude for general $k$ is constructed as [44],

$$
m_n^{(k)}(\alpha|\beta) = \int d\mu_n \mathrm{PT}^{(k)}[\alpha]\mathrm{PT}^{(k)}[\beta],
\tag{3.8}
$$

where $\alpha$ and $\beta$ are specific ordering of $n$ scalars in this amplitude, and PT stands for the Parke-Taylor factor, which for fixed $k$ takes the form,

$$
\mathrm{PT}^{(k)}[1, 2, \cdots, n] := \frac{1}{|1\,2\,\cdots\,k||2\,3\,\cdots\,k-1|\cdots|n\,1\,\cdots\,k-1|}.
\tag{3.9}
$$

Two copies of the PT factors signifies that the scalar is in the biadjoint representation. We will work with $\alpha = \beta = \mathbb{I}$.

In this section, we will study soft theorems for the generalised biadjoint scalars for any $k$. We will begin by recalling the single soft theorem, which was studied in [49]. We will first review the case for $k = 3$ theory and then summarise the general $k$ case briefly. Through this review, we will set up the notation, which will be useful when we study the double soft theorem.

## 3.1 Single soft theorems for $k = 3$ amplitudes

Let us consider the $n$-point amplitude with $n$-th state going soft. A convenient way to deal with the soft limit is to express the $n$-point function with one soft state in terms of $(n-1)$-point function. This is done by extracting the terms in the PT factor which depend on the puncture corresponding to the $n$-th particle. After extracting those terms, the remaining PT factor is almost that for the $n-1$ point function. However, to ensure the cyclic symmetry of the PT factor we need to multiply and divide by terms which reinstate the cyclic symmetry of the $(n-1)$-point PT factor. Taking this into account we can write,

$$
\mathrm{PT}^{(3)}[1, 2, \cdots, n] = \frac{|n-2\,n-1\,1||n-1\,1\,2|}{|n-2\,n-1\,n||n-1\,n\,1||n\,1\,2|}\mathrm{PT}^{(3)}[1, 2, \cdots n-1].
\tag{3.10}
$$

On the RHS of Eq. (3.10), the PT factor involves $n-1$ punctures and the factor multiplying it has two parts, while the denominator contains the soft punctures extracted from the $n$-point PT factor, the numerator terms are remnants of the cyclic symmetry of the $n-1$-point function. In order to take the soft limit, we introduce a parameter $\tau$ and introduce the limit,

$$
s_{nab} = \tau \hat{s}_{nab} \to 0, \qquad \forall a, b \neq n,
\tag{3.11}
$$

where $s_{abc}$ is a symmetric third rank tensor whose components are the generalised Mandelstam variables for $k = 3$ case. The limit taken above is akin to the limit $s_{ab} \to \tau \hat{s}_{ab}$ in the $k = 2$ case where $s_{ab}$ are the Mandelstam variables for $k = 2$ which can be expressed in terms of the kinematic variables.

In this limit scattering equations decouple at the leading order, since $s_{a\,b\,n}$ scales differently compared to $s_{a\,b\,c}$ when $\{a,b,c\} \neq n$. The decoupled equations take the form[1],

$$E_n^{(i)} = \tau \sum_{1 \le a < b \le n-1} \frac{\hat{s}_{a\,b\,n}}{|a\ b\ n|} \frac{\partial}{\partial x_n^i} |a\ b\ n| = 0\,,$$

(3.12)

$$E_a^{(i)} = \sum_{\{b,c\} \neq \{a,n\}} \frac{s_{a\,b\,c}}{|a\ b\ c|} \frac{\partial}{\partial x_a^i} |a\ b\ c| = 0, \qquad \forall a \neq n, \quad i = 1, 2\,.$$

In the soft limit we replace the $n$-point PT factor by the $n-1$-point PT factor multiplied by the soft factor extracted from the original PT factor as given in Eq. (3.10). With this substitution and explicitly pulling out the integration over the soft kinematic variables, Eq. (3.8) takes the form,

$$m_n^{(3)}(\mathbf{I}|\mathbf{I}) = \int d\mu_{n-1} \prod_{i=1}^{2} \int dx_n^i \delta(E_n^{(i)}) \left( \frac{|n-2\ n-1\ 1||n-1\ 1\ 2|}{|n-2\ n-1\ n||n-1\ n\ 1||n\ 1\ 2|} \right)^2$$
$$\times \left( \mathrm{PT}^{(3)}[1, 2, \cdots, n-1] \right)^2$$

(3.13)

$$:= \mathrm{S}_n^{(3)} m_{n-1}^{(3)}(\mathbf{I}|\mathbf{I})\,,$$

where

$$\mathrm{S}_n^{(3)} = \prod_{i=1}^{2} \int dx_n^i \delta(E_n^{(i)}) \left( \frac{|n-2\ n-1\ 1||n-1\ 1\ 2|}{|n-2\ n-1\ n||n-1\ n\ 1||n\ 1\ 2|} \right)^2\,.$$

(3.14)

The integration over $x_n^i$ in Eq. (3.14) implements the scattering equations Eq. (3.12) through the $\delta$-functions. Equivalently, it is convenient to replace the $\delta(E_n^{(i)})$ by poles located at the zeros of the scattering equations. This is achieved by removing the $\delta$-functions and putting the scattering equations in the denominator of the integrand. This allows us to use the contour integral method, and we can compute the integral by deforming the contour away from the zeros of the scattering equation. The residues collected from the poles away from the zeroes of the scattering equation give identical contribution except with the opposite sign. Since we will be working with multiple complex variables, two per puncture in the projective coordinates, as in Eq. (3.13) or three per puncture in the homogeneous coordinates, it is suitable to use the global residue theorem [68] to carry out the contour deformation.

Since the contribution from the scattering equations is not picked up in the contour deformation, only possible contributions come from the soft factors in Eq. (3.14). These contributions can be categorized into two types, collision singularities and collinear singularities. The collision singularities are those where the puncture $\sigma_n$ corresponding to the soft state collides with one of the hard punctures in the denominator of the integrand of Eq. (3.14). The collinear singularities on the other hand, correspond to two punctures becoming collinear with the soft punctures in the denominator of the integrand of Eq. (3.14). We will first discuss the contribution from collision singularities and then discuss the contribution from collinear singularities.

---

[1]In fact, in the $E_a^{(i)}$ equation there is an additional term $\tau \frac{\hat{s}_{a\,b\,n}}{|a\ b\ n|} \frac{\partial}{\partial x_a^i} |a\ b\ n|$. When one considers the singular configuration $|a\ b\ n| \sim \tau$, which can happen when $a, b, n$ are collinear or $n$-th puncture collides with either puncture $a$ or puncture $b$, then the derivative term is of the form $\begin{vmatrix} 1 & 1 \\ x_b^j & x_n^j \end{vmatrix}$, where $j \neq i$. This determinant, in general, may not scale to 0 as $\tau \to 0$. Hence in the case of the singular solution this additional term is of $\mathcal{O}(1)$, nevertheless its contribution appears at subleading order in the soft theorem [67]. We will ignore these configurations as we are interested in the leading order results, which have contributions only from the regular solutions.

### 3.1.1 Collision singularities

The collision singularities in Eq. (3.14) occur for the following two planar configurations: (i) $\sigma_n \to \sigma_1$, (ii) $\sigma_n \to \sigma_{n-1}$. In the first case we parametrize the $n$-th variable as[2],

$$x_n = x_1 + \epsilon, \qquad y_n = y_1 + \epsilon\alpha, \quad \epsilon \to 0, \tag{3.15}$$

where we have chosen to work in the inhomogeneous coordinates. With this parametrisation the integration measure over the soft puncture variables becomes,

$$\prod_{i=1}^{2} dx_n^i = dx_n dy_n = \epsilon d\epsilon d\alpha, \tag{3.16}$$

and delta functions, treated as the top form on $\sigma_n$, transforms in the following way,

$$\delta^{(2)}\left(\partial_{x_n}\mathcal{S}^{(3)}, \partial_{y_n}\mathcal{S}^{(3)}\right) = \epsilon \delta^{(2)}\left(\partial_\epsilon \mathcal{S}^{(3)}, \partial_\alpha \mathcal{S}^{(3)}\right). \tag{3.17}$$

In deriving this relation we have used the fact that the change of variables implemented for the argument of the $\delta$-function generates a term proportional to $\epsilon^{-1}$, and $\delta(ax) = |a|^{-1}\delta(x)$. The scattering equations become,

$$d\mathcal{S}^{(3)} = \tau \sum_{2 \le b \le n-1} \hat{s}_{1\,b\,n}\left(\frac{d\epsilon}{\epsilon} + \frac{d\alpha}{\alpha - \alpha_b}\right)$$

$$\Rightarrow \quad \partial_\epsilon \mathcal{S}^{(3)} = \frac{\tau}{\epsilon}\sum_{a=2}^{n-1}\hat{s}_{1\,a\,n}, \qquad \partial_\alpha \mathcal{S}^{(3)} = \tau \sum_{2 \le b \le n-1}\frac{\hat{s}_{1\,b\,n}}{\alpha - \alpha_b}, \tag{3.18}$$

where, $\alpha_b$ is the $\mathbb{CP}^1$ projection of $\sigma_b$. The soft factor can then be evaluated as,

$$S_n^{(3)\perp}(n;1) = \frac{1}{\tau^2}\oint \frac{\epsilon^2 d\epsilon}{\frac{1}{\epsilon}\sum_{a=2}^{n-1}\hat{s}_{1\,a\,n}}\oint \frac{d\alpha}{\sum_{b=2}^{n-1}\frac{\hat{s}_{1\,b\,n}}{\alpha - \alpha_b}}\left[\frac{\alpha_2 - \alpha_{n-1}}{\epsilon^2(\alpha - \alpha_{n-1})(\alpha - \alpha_2)}\right]^2$$

$$= \frac{1}{\sum_{a=2}^{n-1}s_{1\,a\,n}}\left(\frac{1}{s_{1\,2\,n}} + \frac{1}{s_{n-1\,n\,1}}\right), \tag{3.19}$$

where the integration in $\alpha$ variable is that corresponding to a single soft factor for $k = 2$ from Eq. (2.7). A similar analysis in the case (ii), when $\sigma_n \to \sigma_{n-1}$, leads the corresponding residue to be equal to,

$$S_n^{(3)\perp}(n;n-1) = \frac{1}{\sum_{a=1}^{n-2}s_{n-1\,n\,a}}\left(\frac{1}{s_{n-1\,n\,1}} + \frac{1}{s_{n-2\,n-1\,n}}\right). \tag{3.20}$$

Besides the planar collisions, we could also have non-planar collisions, e.g., $\sigma_n \to \sigma_2$, $\sigma_n \to \sigma_{n-2}$. Most of the analysis above carries through in these cases as well except that in these cases we have only one determinant in the denominator of the soft factor becoming proportional to $\epsilon$. This gives us $\epsilon^2$ factor in the denominator, which is not good enough to offset $\epsilon^3$ factor in the numerator and as a result this type of degeneration does not contribute to the single soft theorem.

---

[2]For the $k = 3$ case, we will use $\{1, x_a^1, x_a^2\}$ or $\{1, x_a, y_a\}$ interchangeably, to denote the inhomogeneous coordinates of the $\sigma_a$ puncture.

### 3.1.2 Collinear singularities

We will now look at the collinear singularities. As mentioned earlier, collinear singularities occur when the soft particle $n$ becomes collinear with two hard punctures. We will begin with the case where the soft puncture of $n$ becomes collinear with hard punctures of $n-2$ and $n-1$. In the homogeneous coordinates, we have,

$$\sigma_n = \alpha \sigma_{n-1} + \xi, \tag{3.21}$$

where $\xi$ lies on the straight line that connects the punctures $n-1$ and $n-2$. In this case, the determinant $(n-2\,n-1\,n)$ becomes,

$$(n-2\,n-1\,n) = (n-2\,n-1\,\xi) = 0, \tag{3.22}$$

where we have used the fact that $\xi$ is collinear with $n-2$ and $n-1$. The Eq. (3.22) corresponds to poles of the PT factor. However note that a soft puncture becoming collinear with two hard punctures is a codimension one singularity. Whereas for the $\mathbb{CP}^2$ integration to give non-zero residue, one needs a codimension two singularity. Such a codimension two singularity occurs when the soft puncture becomes simultaneously collinear with two sets of hard punctures. We will therefore consider the case where soft puncture $n$ becomes collinear with two pairs of hard punctures $n-2, n-1$ and $1, 2$. This can be parametrised as,

$$\sigma_n = \alpha \sigma_{n-1} + \beta \sigma_1 + \xi, \tag{3.23}$$

where, as $\beta \to 0$ we obtain the straight line corresponding to the punctures $n-2, n-1$ and, as $\alpha \to 0$ we obtain the straight line corresponding to the punctures $1, 2$. The above equation when written in inhomogeneous coordinates reads,

$$
\begin{aligned}
x_n &= \alpha x_{n-1} + \beta x_1 + (1-(\alpha+\beta))x_\xi \\
y_n &= \alpha y_{n-1} + \beta y_1 + (1-(\alpha+\beta))y_\xi,
\end{aligned}
\tag{3.24}
$$

which makes the determinants in the PT factor transform as,

$$
\begin{aligned}
|n-2\,n-1\,n| &= \beta|n-2\,n-1\,1|, \\
|n\,1\,2| &= \alpha|n-1\,1\,2|, \\
|n-1\,n\,1| &= |n-1\,\xi\,1|.
\end{aligned}
\tag{3.25}
$$

The measure, on the other hand, transforms as,

$$dx_n dy_n = |n-1\,1\,\xi| d\alpha d\beta, \tag{3.26}$$

and the scattering equations become,

$$
\begin{aligned}
d\mathcal{S}^{(3)} &= \frac{s_{n-2\,n-1\,n}}{\alpha}d\alpha + \frac{s_{n\,1\,2}}{\beta}d\beta \\
\Rightarrow \partial_\alpha \mathcal{S}^{(3)} &= \frac{s_{n-2\,n-1\,n}}{\alpha}, \qquad \partial_\beta \mathcal{S}^{(3)} = \frac{s_{n\,1\,2}}{\beta}.
\end{aligned}
\tag{3.27}
$$

The transformation rule Eq. (3.24) implies,

$$\delta^2\left(\partial_{x_n}\mathcal{S}^{(3)}, \partial_{y_n}\mathcal{S}^{(3)}\right) = |n-1\,1\,\xi|\delta^2\left(\partial_\alpha \mathcal{S}^{(3)}, \partial_\beta \mathcal{S}^{(3)}\right). \tag{3.28}$$

The soft factor then takes the form,

$$\mathrm{S}_n^{(3)\|} = \int \frac{(|n-1\,1\,\xi|)^2 d\alpha d\beta}{\frac{s_{n-2\,n-1\,n}}{\alpha} \times \frac{s_{n12}}{\beta}} \left(\frac{1}{\alpha\beta|n-1\,\xi\,1|}\right)^2$$

$$= \frac{1}{s_{n-2\,n-1\,n}\,s_{n\,1\,2}}. \tag{3.29}$$

Therefore the full single soft factor for $k = 3$ biadjoint scalar theory is given by,

$$
\begin{aligned}
S_n^{(3)} &= S_n^{(3)\perp}(n;1) + S_n^{(3)\perp}(n-1;n) + S_n^{(3)\parallel} \\
&= \frac{1}{\displaystyle\sum_{a=2}^{n-1} s_{1\,a\,n}}\left(\frac{1}{s_{1\,2\,n}} + \frac{1}{s_{n-1\,n\,1}}\right) + \frac{1}{\displaystyle\sum_{a=1}^{n-2} s_{n-1\,n\,a}}\left(\frac{1}{s_{n-1\,n\,1}} + \frac{1}{s_{n-2\,n-1\,n}}\right) \\
&\qquad\qquad + \frac{1}{s_{n-2\,n-1\,n}\,s_{n12}}\,.
\end{aligned}
\tag{3.30}
$$

Thus the single soft factor for $k = 3$ is obtained by studying the boundary structure in the moduli space which comprises of both collision and collinear type singularities of codimension two.

## 3.2   Single soft limit for arbitrary $k$

The above discussions for single soft theorem can be generalised for arbitrary $k$. We consider soft limit in $n$-th external state, such that $s_{a_1 a_2 \cdots a_{k-1} n}$ scales as $\tau$ with $\tau \to 0$ for any $a_i \in \{1, 2, \cdots n-1\}$. The scattering equations can then be decomposed in the following way:

$$
\begin{aligned}
E_{a_1}^{(i)} &= \sum_{\substack{1\le a_2 < a_3 < \cdots < a_k \le n-1 \\ a_2,\cdots, a_k \ne a_1}} \frac{s_{a_1 a_2 \cdots a_k}}{|a_1\,a_2\,\cdots\,a_k|}\frac{\partial}{\partial x_{a_1}^i}|a_1\,a_2\,\cdots\,a_k| = 0, \qquad \forall a_1 \\
E_n^{(i)} &= \sum_{1\le a_1 < \cdots < a_{k-1}\le n-1} \frac{s_{a_1 \cdots a_{k-1} n}}{|a_1\,\cdots\,a_{k-1}\,\sigma_n|}\frac{\partial}{\partial x_n^i}|a_1\,\cdots\,a_{k-1}\,\sigma_n| = 0,
\end{aligned}
\tag{3.31}
$$

where $i = 1, 2, \cdots k-1$. Here we consider only the regular solutions to scattering equations and hence we neglect any $\mathcal{O}(1)$ terms that may arise when $|a_1 \cdots a_{k-1}\sigma_n| \sim \tau$.

The soft factor for arbitrary $k$ can be expressed in the integral form as,

$$
\begin{aligned}
S^{(k)} &= \int \prod_{i=1}^{k-1} dx_n^i\,\delta^{(k-1)}\!\left(E_n^i\right) \\
&\quad \times \left[\frac{|(n-k+1)\cdots 1||(n-k+2)\cdots 1\,2|\cdots|n-1\,1\,2\cdots k-1|}{|(n-k+1)\cdots n||(n-k+2)\cdots n\,1|\cdots|n\,1\,2\cdots k-1|}\right]^2.
\end{aligned}
\tag{3.32}
$$

In [49], the expression in Eq. (3.32) was evaluated in terms of the generalised Mandelstam variables by an iterative procedure and a prescription for calculating the soft factor for any given $k$ was presented. The scaling of the soft factor in this case can be seen to be $\tau^{-(k-1)}$. For the purposes of this work, the formal factorisation given in Eq. (3.32) is sufficient and we refer the reader to [49] for further details on the single soft factor.

# 4   Double Soft Theorem for $k = 3$ Amplitudes

We will now look at the double soft theorems, where we have two external states becoming soft. Here we have two main cases, and we will treat them one at a time. To begin with, we will discuss the simultaneous double soft limit. In this case, both the external states are going soft at the same rate, and therefore the limits cannot be taken independently. The simultaneous

double soft theorem has multiple sub-cases, and we will analyse each sub-case separately. The other kind of double soft limit is called the consecutive double soft limit, where one state goes soft at a faster rate than the other soft state. This limit clearly has a hierarchical structure and is relatively easy to deal with, and we will take up this case later.

## 4.1 Simultaneous double soft limit

Let us now look at the simultaneous double soft limit. This limit implies two external states are going soft at the same rate, parametrised in terms of $\tau$, which in the soft limit will be taken to zero. In this case, we can have several configurations which qualify as double soft limit, but their treatment differs. For example, we can have adjacent external states going soft, which is the case we will deal with first. However, besides this, we can have next to adjacent states becoming soft. The singularity structure of this case is quite different from the case of adjacent external states going soft. We could also have the next to next to adjacent states in the scattering going soft. For the case at hand, namely $k = 3$, this choice of configurations with soft scattering states further away from each other, resulting in an expression which is a product of two single soft factors. In fact, for arbitrary $k$, if there are at least $k-1$ hard states between two soft states, then the double soft limit is just a product of two single soft factors.

**Soft Limits for Adjacent Particles:** We will consider $n$-th and $(n-1)$-th external states going soft. This limit corresponds to the following behaviour of the generalised Mandelstam variables,

$$s_{a\,b\,n-1} \to \tau \hat{s}_{a\,b\,n-1}, \quad s_{a\,b\,n} \to \tau \hat{s}_{a\,b\,n}, \quad s_{a\,n-1\,n} \to \tau^2 \hat{s}_{a\,n-1\,n}, \quad a, b = 1, 2, \cdots, n-2. \quad (4.1)$$

Here we will be concerned with the leading order soft factorisation. Therefore it suffices to consider only regular solutions to the scattering equations, *i.e.*, we assume none of the determinants $|a\,b\,n|$ and $|a\,b\,n-1|$ scale as $\mathcal{O}(\tau) \,\forall a, b \in \{1, 2, \cdots, n-2\}$. In [67] it has been shown that singular solutions to scattering equations for $k \geq 3$ contribute to the subleading soft theorem. Therefore these solutions will not be part of our analysis. The scattering equations to the leading order can be written as,

$$E_a^{(i)} = \sum_{b,c \neq a, n-1, n} \frac{s_{a\,b\,c}}{|a\,b\,c|} \frac{\partial}{\partial x_a^{(i)}} |a\,b\,c| = 0, \qquad \forall a$$

$$E_{n-1}^{(i)} = \tau \sum_{a,b \neq n-1, n} \frac{\hat{s}_{a\,b\,n-1}}{|a\,b\,n-1|} \frac{\partial}{\partial x_{n-1}^{(i)}} |a\,b\,n-1| + \tau^2 \sum_{a=1}^{n-2} \frac{\hat{s}_{a\,n-1\,n}}{|a\,n-1\,n|} \frac{\partial}{\partial x_{n-1}^{(i)}} |a\,n-1\,n| = 0,$$

$$E_n^{(i)} = \tau \sum_{a,b \neq n-1, n} \frac{\hat{s}_{a\,b\,n}}{|a\,b\,n|} \frac{\partial}{\partial x_n^{(i)}} |a\,b\,n| + \tau^2 \sum_{a=1}^{n-2} \frac{\hat{s}_{a\,n-1\,n}}{|a\,n-1\,n|} \frac{\partial}{\partial x_n^{(i)}} |a\,n-1\,n| = 0, \qquad (4.2)$$

where $i = 1, 2$ labels components of the inhomogeneous coordinates of the puncture.

Let us first look at the integrand in Eq. (3.8) with $(n-1)$-th and $n$-th particle soft. In the double soft limit, as was done in the single soft limit, we extract the dependence on the soft punctures and write the remaining factor as the Parke-Taylor factor for $n-2$ punctures. The resulting expression can be written as,

$$\text{PT}^{(3)}[12 \cdots n] = \frac{|n-3\,n-2\,1||n-2\,1\,2|}{|n-3\,n-2\,n-1||n-2\,n-1\,n||n-1\,n\,1||n\,1\,2|} \text{PT}^{(3)}[12 \cdots n-2]. \quad (4.3)$$

In the double soft limit, the amplitude can be written as,

$$m_n^{(3)}(\text{I}|\text{I}) = m_{n-2}^{(3)}(\text{I}|\text{I}) \int \prod_i dx_{n-1}^{(i)} dx_n^{(i)} \delta(E_{n-1}^{(i)}) \delta(E_n^{(i)})$$

$$\times \left( \frac{|n-3\ n-2\ 1||n-2\ 1\ 2|}{|n-3\ n-2\ n-1||n-2\ n-1\ n||n-1\ n\ 1||n\ 1\ 2|} \right)^2 . \quad (4.4)$$

The soft factor can be computed by deforming contours of integration away from the original poles coming from scattering equations of soft particles which are written in terms of the delta-functions in the integral. In this process, we encounter poles in the integrand, which occur when the determinants in the denominator of the integrand vanish. As in the single soft limit, the determinant vanishes in two possible ways. Either there is a collision singularity, that is when two punctures collide or when three punctures become collinear. In the case of the double soft limit, we encounter more intricate combinations of these two types of singularities.

The second and third equations in Eq. (4.2) have two sums. First sum always scales linearly in $\tau$, however, depending on the behaviour of $|a\ n-1\ n|$, the terms in the second sum can either be linear or quadratic in $\tau$. This gives rise to two types of solutions to the scattering equations:

- Non-degenerate solutions: They correspond to $|a\ n-1\ n| \sim \mathcal{O}(\tau^0)$. In this case, the second sum of last two equations in Eq. (4.2) are sub-dominant compared to the first sum.

- Degenerate solutions: They correspond to $|a\ n-1\ n| \sim \mathcal{O}(\tau)$. In this case, some or all the terms in the second sum are of $\mathcal{O}(\tau)$ and hence are of the same order as the first sum.

For non-degenerate solutions the double soft factor scales as $\tau^{-4}$. This behaviour can be understood from Eq.(4.4) - there are four delta functions containing scattering equations of soft external states in the arguments, each of which contributes to a factor of $\tau$ in the denominator, and the integrand is independent of $\tau$. For degenerate solutions, the determinants which depend on both $n-1$ and $n$ punctures contribute to a factor of $\tau$ each, and there are two such determinants in the denominator of the integrand. So the integrand scales as $\tau^{-4}$, and it can be checked that measure goes as $\tau^{-2}$ making the overall scaling of the double soft factor as $\tau^{-6}$. As we are interested in the leading soft theorem ,we will only present the analysis of the degenerate solutions in the following subsection.

## 4.2 Degenerate solutions

As argued above, similar to $k = 2$ case studied in sec.(2.2), degenerate solutions dominate in the double soft limit for the adjacent particles going soft. We will therefore concentrate on this sector and analyse the leading singular behaviour in the double soft limit. In order to do that, let us first look at the scattering equations in Eq.(4.2). For the degenerate solutions denominator in the second sum of $E_{n-1}$ and $E_n$ equations scale as $\tau$, making the overall scaling of the terms to be $\tau$. There are two possible ways where the determinant $|a\ n-1\ n| \sim \tau$:

- if the two soft punctures collide with each other, $\sigma_n - \sigma_{n-1} \sim \tau$.

- if the two soft punctures are nearly collinear to any one of the hard punctures. This can happen if the straight line joining $\sigma_{n-1}$ and $\sigma_n$ is away from a hard puncture, $\sigma_a$ by a distance of $\mathcal{O}(\tau)$.[3]

---

[3]There can not be more than two such hard punctures because in that case the two hard punctures, say, $a$ and $b$ will be collinear with a soft puncture in the limit $\tau \to 0$ making both $|a\ n-1\ n| \to 0$ and $|b\ n-1\ n| \to 0$. But in the latter case, they are part of singular solutions and hence we do not consider them in this discussion.

We will analyse these two cases in detail in the following subsections.

### 4.2.1 Collision of soft punctures

We will begin with the analysis of the collision singularity. It is convenient to make the following change of variables in order to do the integration:

$$x_{n-1}^i = \rho^i + \xi^i, \qquad x_n^i = \rho^i - \xi^i, \qquad i = 1, 2, \tag{4.5}$$

where $\rho$ is $\mathcal{O}(1)$ and $\xi$ is $\mathcal{O}(\tau)$. In component form they can be given by $\rho = \begin{pmatrix} 1 \\ \rho^1 \\ \rho^2 \end{pmatrix}$ and

$\xi = \begin{pmatrix} 0 \\ \xi^1 \\ \xi^2 \end{pmatrix}$. In terms of these variables we can re-express the scattering equations as,

$$E_{n-1}^{(i)} + E_n^{(i)} = \sum_{1 \le a < b \le n-2} \frac{s_{a\,b\,n-1} + s_{a\,b\,n}}{|a\,b\,\rho|} \frac{\partial}{\partial \rho^i} |a\,b\,\rho|,$$

$$E_{n-1}^{(i)} - E_n^{(i)} = \sum_{1 \le a < b \le n-2} \frac{s_{a\,b\,n-1} - s_{a\,b\,n}}{|a\,b\,\rho|} \frac{\partial}{\partial \rho^i} |a\,b\,\rho| + \sum_{a=1}^{n-2} \frac{s_{a\,n-1\,n}}{|a\,\rho\,\xi|} \frac{\partial}{\partial \xi^i} |a\,\rho\,\xi|. \tag{4.6}$$

Determinant containing $a, \rho$ and $\xi$ then takes the form,

$$|a\,\rho\,\xi| = \begin{vmatrix} 1 & 1 & 0 \\ x_a^1 & \rho^1 & \xi^1 \\ x_a^2 & \rho^2 & \xi^2 \end{vmatrix}$$

$$= \xi^1(\rho^1 - x_a^1)(\alpha - \alpha_a), \qquad \text{where we define} \quad \alpha = \frac{\xi^2}{\xi^1}, \quad \alpha_a = \frac{x_a^2 - \rho^2}{x_a^1 - \rho^1}. \tag{4.7}$$

The measure transforms as,

$$\prod_{i=1}^2 dx_{n-1}^i dx_n^i \delta(E_{n-1}^{(i)}) \delta(E_n^{(i)}) = 16\, d^2\rho\, d^2\xi\, \delta^{(2)}(E_{n-1} + E_n) \delta^{(2)}(E_{n-1} - E_n). \tag{4.8}$$

The second delta function in the measure can be used to solve for $\xi$ in terms of $\rho$ to localize the $\xi$ integration analogous to the $k = 2$ case [54–58],

$$\delta^{(2)}(E_{n-1} - E_n) = \sum_{\xi_0} \frac{\delta^{(2)}(\xi - \xi_0)}{\begin{vmatrix} \frac{\partial(E_{n-1}^{(1)} - E_n^{(1)})}{\partial \xi^1} & \frac{\partial(E_{n-1}^{(1)} - E_n^{(1)})}{\partial \xi^2} \\ \frac{\partial(E_{n-1}^{(2)} - E_n^{(2)})}{\partial \xi^1} & \frac{\partial(E_{n-1}^{(2)} - E_n^{(2)})}{\partial \xi^2} \end{vmatrix}}, \tag{4.9}$$

where $\xi_0$ are the solutions of the scattering equations for $\xi$.

However, we find that solving for $\xi$ from the scattering equations is rather complicated as it leads to a high degree polynomial equation. Therefore we take an alternate approach analogous to the one we adopted while discussing $k = 2$ soft limit in Sec.(2.2).

Instead of using the second delta function to localize $\xi$ integral, we will use it to convert the $\xi$ integration to a contour integral. We can express the soft factor as,

$$S_{\text{deg}}^{(3)} = \int d^2\rho\, d^2\xi\, \delta^{(2)}(E_{n-1} + E_n) \delta^{(2)}(E_{n-1} - E_n) \left[ \frac{|n-3\,n-2\,1| |n-2\,1\,2|}{|n-3\,n-2\,\rho| |n-2\,\xi\,\rho| |\rho\,\xi\,1| |\rho\,1\,2|} \right]^2. \tag{4.10}$$

In $S_{\text{deg}}^{(3)}$, the subscript implies that the soft factor is related to the degenerate solutions. We note that,

$$|n-2\,\xi\,\rho||\rho\,\xi\,1| = (\xi^1)^2(\alpha-\alpha_{n-2})(\alpha-\alpha_1)(x_{n-2}^1-\rho^1)(x_1^1-\rho^1)\,. \qquad (4.11)$$

Vanishing of the L.H.S. implies either $\xi^1 \to 0$, and/or $\alpha \to \alpha_{n-1}$, and/or $\alpha \to \alpha_1$. We consider the following change of variables,

$$\begin{aligned} \xi^1 &= \epsilon \\ \Rightarrow \xi^2 &= \epsilon\alpha\,, \end{aligned} \qquad (4.12)$$

which simplifies the integration measure,

$$d^2\xi = \begin{vmatrix} 1 & 0 \\ \alpha & \epsilon \end{vmatrix} d\epsilon\,d\alpha = \epsilon\,d\epsilon\,d\alpha\,. \qquad (4.13)$$

Since $\xi^1$ and $\xi^2$ are components of a two dimensional vector, contour deformations for them cannot be done independently because the original contour wraps around the solutions of the scattering equations.

We note that in the limit $\epsilon \to 0$, the dominating term is the second summation in the second and third equations of (4.6). We can, therefore, neglect the $\rho$ dependent part, and write the last two equations as,

$$E_{n-1}^{(i)} - E_n^{(i)} = \frac{\partial}{\partial\xi^i}\tilde{S}^3, \qquad \text{where} \quad \tilde{S}^3 = \sum_{a=1}^{n-2} s_{a\,n-1\,n}\log|a\,\rho\,\xi|\,. \qquad (4.14)$$

Thus we have,

$$\begin{pmatrix} \frac{\partial\tilde{S}^3}{\partial\xi^1} \\ \frac{\partial\tilde{S}^3}{\partial\xi^2} \end{pmatrix} = \begin{pmatrix} 1 & -\frac{1}{\epsilon\alpha} \\ 0 & \frac{1}{\epsilon} \end{pmatrix} \begin{pmatrix} \frac{\partial\tilde{S}^3}{\partial\epsilon} \\ \frac{\partial\tilde{S}^3}{\partial\alpha} \end{pmatrix}$$

$$\Rightarrow \quad \delta^{(2)}\left(\frac{\partial\tilde{S}^3}{\partial\xi^1}, \frac{\partial\tilde{S}^3}{\partial\xi^2}\right) = \epsilon\,\delta^{(2)}\left(\frac{\partial\tilde{S}^3}{\partial\epsilon}, \frac{\partial\tilde{S}^3}{\partial\alpha}\right)\,. \qquad (4.15)$$

To see the factorisation channels we write,

$$d\tilde{S}^3 = \sum_{a=1}^{n-2} s_{a\,n-1\,n}\left[\frac{d\epsilon}{\epsilon} + \frac{d\alpha}{\alpha-\alpha_a}\right], \qquad \alpha_a = \frac{x_a^2-\rho^2}{x_a^1-\rho^1}\,, \qquad (4.16)$$

and find poles are at $\epsilon=0$ and $\alpha=\alpha_1,\alpha_{n-2}$. This is evident from the $\xi$ integration which follows from Eq.(4.10),

$$\oint \frac{\epsilon^2 d\epsilon\,d\alpha}{\frac{1}{\epsilon}\sum_{a=1}^{n-2} s_{a\,n-1\,n}\sum_{a=1}^{n-2}\frac{s_{a\,n-1\,n}}{\alpha-\alpha_a}}\left[\frac{1}{\underline{\epsilon^2}(x_1^1-\rho^1)(x_{n-2}^1-\rho^1)\underline{(\alpha-\alpha_1)(\alpha-\alpha_{n-2})}}\right]^2$$

$$= \frac{1}{\sum_{a=1}^{n-2} s_{a\,n-1\,n}}\left[\frac{1}{s_{1\,n-1\,n}} + \frac{1}{s_{n-2\,n-1\,n}}\right]\frac{1}{|1\,n-2\,\rho|^2}\,. \qquad (4.17)$$

Now we have to perform $\rho$ integration,

$$S_{\text{deg}}^{(3)} = \frac{1}{\sum_{a=1}^{n-2} s_{a\,n-1\,n}}\left[\frac{1}{s_{1\,n-1\,n}} + \frac{1}{s_{n-2\,n-1\,n}}\right]$$

$$\times \int d^2\rho \; \delta^{(2)}(E_{n-1} + E_n) \left[ \frac{|n-3\,n-2\,1||n-2\,1\,2|}{|n-3\,n-2\,\rho||1\,n-2\,\rho||\rho\,1\,2|} \right]^2 . \tag{4.18}$$

The $\rho$ integration is similar to the single soft analysis with the generalised potential function $\tilde{S}'^{(3)} = \sum_{1 \le a < b \le n-2} (s_{a\,b\,n-1} + s_{a\,b\,n}) \log |a\,b\,\rho|$. We therefore obtain,

$$S_{\text{deg}}^{(3)} = \frac{1}{\sum\limits_{a=1}^{n-2} s_{a\,n-1\,n}} \left( \frac{1}{s_{n-1\,n\,1}} + \frac{1}{s_{n-2\,n-1\,n}} \right)$$

$$\times \left[ \frac{1}{(s_{n-3\,n-2\,n-1} + s_{n-3\,n-2\,n})(s_{n-1\,1\,2} + s_{n\,1\,2})} \right.$$

$$+ \frac{1}{\sum\limits_{a=1}^{n-3} (s_{a\,n-2\,n-1} + s_{a\,n-2\,n})} \left( \frac{1}{s_{n-3\,n-2\,n-1} + s_{n-3\,n-2\,n}} + \frac{1}{s_{n-2\,n-1\,1} + s_{n-2\,n\,1}} \right)$$

$$+ \left. \frac{1}{\sum\limits_{a=2}^{n-2} (s_{a\,n-1\,1} + s_{a\,n\,1})} \left( \frac{1}{s_{n-2\,n-1\,1} + s_{n-2\,n\,1}} + \frac{1}{s_{n-1\,1\,2} + s_{n\,1\,2}} \right) \right] . \tag{4.19}$$

#### 4.2.2 Soft punctures collinear to one hard puncture

We will consider $\sigma_{n-1}$ and $\sigma_n$ to be nearly collinear with a puncture $\sigma_d$ corresponding to a hard external state, such that $|\sigma_d\,\sigma_{n-1}\,\sigma_n| \sim \tau$. It can then be seen from Eq.(4.4) that integrand for the double soft factor goes as $\tau^{-2}$ whenever $d = n-2$ or 1. For other values of $d$ no determinant in the denominator becomes of $\mathcal{O}(\tau)$, therefore, this configuration produces subleading contribution compared to the case when two soft punctures collide.

Hence the leading double soft factor $S_{\text{DS}}^{(3)}$ for the adjacent labels, $n-1$ and $n$ going soft simultaneously is given by the expression in Eq.(4.19).

### 4.3 Consecutive double soft limit

We will now look at the consecutive double soft theorem, that is, where one external particle becomes soft at a faster rate than the other external soft particle. Here we have two possibilities,

**a)** We will first take the $n$-th particle to be soft, *i.e.*, $s_{n\,a\,b} = \tau_1 \hat{s}_{n\,a\,b}$ and we know from the single soft theorem for $k = 3$ that,

$$m_n^{(3)}(\mathbb{I}|\mathbb{I}) = S_{n\to 0}^{(3)} m_{n-1}^{(3)}(\mathbb{I}|\mathbb{I}) , \tag{4.20}$$

with the soft factor,

$$S_{n\to 0}^{(3)} = \frac{1}{\tau_1^2} \left[ \frac{1}{\hat{s}_{n-2\,n-1\,n}\hat{s}_{n\,1\,2}} + \frac{1}{\sum\limits_{a=2}^{n-1} \hat{s}_{a\,n\,1}} \left( \frac{1}{\hat{s}_{n-1\,n\,1}} + \frac{1}{\hat{s}_{n\,1\,2}} \right) \right.$$

$$\left. + \frac{1}{\sum\limits_{a=1}^{n-2} \hat{s}_{a\,n-1\,n}} \left( \frac{1}{\hat{s}_{n-1\,n\,1}} + \frac{1}{\hat{s}_{n-2\,n-1\,n}} \right) \right] . \tag{4.21}$$

We follow it up by considering the soft limit for the $(n-1)$-th particle, $s_{n-1\,a\,b} = \tau_2 \hat{s}_{n-1\,a\,b}$, with the condition $\tau_1 \ll \tau_2$. Therefore we obtain,

$$m_n^{(3)}(\mathrm{I}|\mathrm{I}) = \mathrm{S}_{n\to0}^{(3)}\Big|_{n-1\to0} \mathrm{S}_{n-1\to0}^{(3)} m_{n-2}^{(3)}(\mathrm{I}|\mathrm{I}), \qquad (4.22)$$

where the second soft factor $\mathrm{S}_{n-1\to0}^{(3)}$ has the following form,

$$\mathrm{S}_{n-1\to0}^{(3)} = \frac{1}{\tau_2^2}\Bigg[ \frac{1}{\hat{s}_{n-3\,n-2\,n-1}\hat{s}_{n-1\,1\,2}} + \frac{1}{\displaystyle\sum_{a=1}^{n-3}\hat{s}_{a\,n-2\,n-1}}\left(\frac{1}{\hat{s}_{n-2\,n-1\,1}} + \frac{1}{\hat{s}_{n-3\,n-2\,n-1}}\right)$$

$$+ \frac{1}{\displaystyle\sum_{a=2}^{n-2}\hat{s}_{a\,n-1\,1}}\left(\frac{1}{\hat{s}_{n-1\,1\,2}} + \frac{1}{\hat{s}_{n-2\,n-1\,1}}\right)\Bigg]. \qquad (4.23)$$

In the limit $\tau_1 \ll \tau_2$, the first soft factor $\mathrm{S}_{n\to0}^{(3)}$ in the leading order of $\tau_1$ as well as of $\tau_2$ takes the form,

$$\mathrm{S}_{n\to0}^{(3)}\Big|_{n-1\to0} = \frac{1}{\tau_1^2}\Bigg[ \frac{1}{\tau_2 \hat{s}_{n-2\,n-1\,n}\hat{s}_{n\,1\,2}} + \frac{1}{\displaystyle\sum_{a=2}^{n-1}\hat{s}_{a\,n\,1}}\left(\frac{1}{\tau_2 \hat{s}_{n-1\,n\,1}} + \frac{1}{\hat{s}_{n\,1\,2}}\right)$$

$$+ \frac{1}{\tau_2\displaystyle\sum_{a=1}^{n-2}\hat{s}_{a\,n-1\,n}}\left(\frac{1}{\tau_2 \hat{s}_{n-1\,n\,1}} + \frac{1}{\tau_2 \hat{s}_{n-2\,n-1\,n}}\right)\Bigg]$$

$$= \frac{1}{\tau_1^2\tau_2^2}\Bigg[ \frac{1}{\displaystyle\sum_{a=1}^{n-2}\hat{s}_{a\,n-1\,n}}\left(\frac{1}{\hat{s}_{n-1\,n\,1}} + \frac{1}{\hat{s}_{n-2\,n-1\,n}}\right)\Bigg]. \qquad (4.24)$$

Thus the consecutive double soft factor for the adjacent particles where $n$-th particle is taken to be softer than the $(n-1)$-th soft particle is given by,

$$\mathrm{S}_{n\to0}^{(3)}\Big|_{n-1\to0} \mathrm{S}_{n-1\to0}^{(3)} = \frac{1}{\tau_1^2\tau_2^4}\Bigg[ \frac{1}{\displaystyle\sum_{a=1}^{n-2}\hat{s}_{a\,n-1\,n}}\left(\frac{1}{\hat{s}_{n-1\,n\,1}} + \frac{1}{\hat{s}_{n-2\,n-1\,n}}\right)\Bigg]$$

$$\times \Bigg[ \frac{1}{\hat{s}_{n-3\,n-2\,n-1}\,\hat{s}_{n-1\,1\,2}} + \frac{1}{\displaystyle\sum_{a=1}^{n-3}\hat{s}_{a\,n-2\,n-1}}\left(\frac{1}{\hat{s}_{n-3\,n-2\,n-1}} + \frac{1}{\hat{s}_{n-2\,n-1\,1}}\right)$$

$$+ \frac{1}{\displaystyle\sum_{a=2}^{n-2}\hat{s}_{a\,n-1\,1}}\left(\frac{1}{\hat{s}_{n-2\,n-1\,1}} + \frac{1}{\hat{s}_{n-1\,1\,2}}\right)\Bigg]. \qquad (4.25)$$

**b)** The second possibility of the consecutive adjacent double soft limit is where we will take the $(n-1)$-th particle to be softer than the $n$-th one, *i.e.*, $\tau_1 \gg \tau_2$. The analysis is similar to the previous one and the soft factorisation turns out to be,

$$m_n^{(3)}(\mathrm{I}|\mathrm{I}) = \mathrm{S}_{n-1\to0}^{(3)}\Big|_{n\to0} \mathrm{S}_{n\to0}^{(3)} m_{n-2}^{(3)}(\mathrm{I}|\mathrm{I}), \qquad (4.26)$$

where the soft factor is given by,

$$
\left.S_{n\to 0}^{(3)}\right|_{n-1\to 0} S_{n-1\to 0}^{(3)} = \frac{1}{\tau_1^4 \tau_2^2}\left[\frac{1}{\displaystyle\sum_{a=1}^{n-2}\hat{s}_{a\,n-1\,n}}\left(\frac{1}{\hat{s}_{n-1\,n\,1}}+\frac{1}{\hat{s}_{n-2\,n-1\,n}}\right)\right]
$$
$$
\times\left[\frac{1}{\hat{s}_{n-3\,n-2\,n}\hat{s}_{n\,1\,2}}+\frac{1}{\displaystyle\sum_{a=1}^{n-3}\hat{s}_{a\,n-2\,n}}\left(\frac{1}{\hat{s}_{n-3\,n-2\,n}}+\frac{1}{\hat{s}_{n-2\,n\,1}}\right)\right.
$$
$$
\left.+\frac{1}{\displaystyle\sum_{a=2}^{n-2}\hat{s}_{a\,n\,1}}\left(\frac{1}{\hat{s}_{n-2\,n\,1}}+\frac{1}{\hat{s}_{n\,1\,2}}\right)\right]. \tag{4.27}
$$

We can obtain Eq.(4.25) and Eq.(4.27) by taking appropriate consecutive limits in $\tau_1$ and $\tau_2$ starting from the expression of simultaneous double soft limit derived in Eq.(4.19). This serves as a consistency check for the simultaneous double soft factor.

## 5 Double Soft Theorem for Arbitrary $k$

The result of sec:(4.1) can be generalised for arbitrary $k$. Contributions for the leading simultaneous double soft factor when two adjacent particles are taken to be soft come from degenerate solutions, more specifically when the two corresponding soft punctures are infinitesimally close to each other, *i.e.*, when the separation is of $\mathcal{O}(\tau)$. The other degenerate configuration occurs when one of the soft punctures approaches the co-dimension one subspace generated by other soft puncture and $(k-2)$ number of hard punctures. This degeneration, however, contributes at subleading order in the soft theorem. Needless to say, the non-degenerate solutions appear at further subleading orders in the expansion and therefore we do not discuss them here. In this section, we present only the leading order result for the double soft theorem in adjacent simultaneous soft limit for any $k \geq 3$.

We will now consider the soft limit in labels $n-1$ and $n$. We can then impose following conditions on the generalised Mandelstam variables:

$$
\begin{aligned}
s_{n-1\,a_1\cdots a_{k-1}} &= \tau\hat{s}_{n-1\,a_1\cdots a_{k-1}}\,,\\
s_{n\,a_1\cdots a_{k-1}} &= \tau\hat{s}_{n\,a_1\cdots a_{k-1}}\,,\\
s_{n-1\,n\,a_1\cdots a_{k-2}} &= \tau^2\hat{s}_{n-1\,n\,a_1\cdots a_{k-2}}\,,
\end{aligned}\tag{5.1}
$$

for $a_1,\cdots a_{k-1}\in\{1,2,\cdots n-2\}$ labelling the hard external states, and rest of the Mandelstam variables are of order unity.

We are interested in the soft limit $m_n^{(k)}(\mathrm{I}|\mathrm{I})=S_{\mathrm{DS}}^{(k)}\, m_{n-2}^{(k)}(\mathrm{I}|\mathrm{I})$ for $k\geq 3$, where the double soft factor can be expressed in the integral form as,

$$
S_{\mathrm{DS}}^{(k)} = \int d^{k-1}x_n\,\delta^{(k-1)}(E_n)\int d^{k-1}x_{n-1}\,\delta^{(k-1)}(E_{n-1})\times
$$
$$
\left[\frac{|n-k\cdots 1||(n-k+1)\cdots 1\,2|\cdots|n-2\,1\cdots k-1|}{|n-k\cdots n-2\,\sigma_{n-1}||(n-k+1)\cdots\sigma_{n-1}\sigma_n|\cdots|\sigma_{n-1}\sigma_n\,1\cdots k-2||\sigma_n\,1\cdots k-1|}\right]^2.
$$
$$\tag{5.2}$$

In the soft limit we can write the scattering equations as follows:

$$
E_{a_1}^{(i)} = \sum_{1\leq a_2\cdots<a_k\leq n-2}\frac{s_{a_1\cdots a_k}}{|a_1\cdots a_k|}\frac{\partial}{\partial x_{a_1}^i}|a_1\cdots a_k|=0,\qquad\forall a_1\in\{1,2,\cdots n-2\}
$$

$$
\begin{aligned}
E_{n-1}^{(i)} &= \sum_{1 \le a_1 \cdots < a_{k-1} \le n-2} \frac{s_{a_1 \cdots a_{k-1}\, n-1}}{|a_1 \cdots a_{k-1}\, \sigma_{n-1}|} \frac{\partial}{\partial x_{n-1}^i} |a_1 \cdots a_{k-1}\, \sigma_{n-1}| \\
&\quad + \sum_{1 \le a_1 \cdots < a_{k-2} \le n-2} \frac{s_{a_1 \cdots a_{k-2}\, n-1\, n}}{|a_1 \cdots a_{k-2}\, \sigma_{n-1}\, \sigma_n|} \frac{\partial}{\partial x_{n-1}^i} |a_1 \cdots a_{k-2}\, \sigma_{n-1}\, \sigma_n| = 0 \,, \\
E_n^{(i)} &= \sum_{1 \le a_1 \cdots < a_{k-1} \le n-2} \frac{s_{a_1 \cdots a_{k-1}\, n}}{|a_1 \cdots a_{k-1}\, \sigma_n|} \frac{\partial}{\partial x_n^i} |a_1 \cdots a_{k-1}\, \sigma_n| \\
&\quad + \sum_{1 \le a_1 \cdots < a_{k-2} \le n-2} \frac{s_{a_1 \cdots a_{k-2}\, n-1\, n}}{|a_1 \cdots a_{k-2}\, \sigma_{n-1}\, \sigma_n|} \frac{\partial}{\partial x_n^i} |a_1 \cdots a_{k-2}\, \sigma_{n-1}\, \sigma_n| = 0 \,, \text{(5.3)}
\end{aligned}
$$

where $i = 1, 2, \cdots k-1$. We would again like to emphasise that here we are only considering the regular solutions and hence have ignored any additional $\mathcal{O}(\tau^0)$ terms in $E_a^{(i)}$ which will be present if we include the singular solutions. To be precise we assume the condition that $|a_1 \cdots a_{k-1}\sigma_{n-1}|$ and $|a_1 \cdots a_{k-1}\sigma_n|$ are always of order one.

We will now focus on the degenerate solutions, *i.e.*, when $|a_1 \cdots a_{k-2}\sigma_{n-1}\sigma_n|$ is of $\mathcal{O}(\tau)$. There are two way to arrive this condition, when $\sigma_{n-1}$ and $\sigma_n$ are in $\mathcal{O}(\tau)$ neighbourhood of each other or when the two soft punctures and any set of $(k-2)$ hard punctures $a_j$, $j \in \{1, 2, \cdots n-2\}$ form a co-dimension one subspace up to $\mathcal{O}(\tau)$ deformation. However, following our analysis earlier, we will ignore the latter configuration.

We choose the following change of variables,

$$
\begin{aligned}
x_{n-1}^i &= \rho^i + \xi^i \,, \\
x_n^i &= \rho^i - \xi^i \,, \qquad \xi^i \sim \mathcal{O}(\tau), \quad i = 1, \cdots k-1 \,.
\end{aligned} \tag{5.4}
$$

The integration measure transforms in the following way,

$$
d^{k-1} x_{n-1}\, d^{k-1} x_n = 2^{k-1} d^{k-1}\rho\, d^{k-1}\xi \,, \tag{5.5}
$$

and the delta functions with arguments as $E_n$ and $E_{n-1}$ are expressed as,

$$
\delta^{(k-1)}(E_n)\, \delta^{(k-1)}(E_{n-1}) = 2^{k-1}\delta^{(k-1)}(E_{n-1} + E_n)\, \delta^{(k-1)}(E_{n-1} - E_n) \,. \tag{5.6}
$$

For later purpose we define,

$$
\begin{aligned}
|a_1 \cdots a_{k-2}\, \sigma_{n-1}\, \sigma_n| &= -2
\begin{vmatrix}
1 & 1 & \cdots & 1 & 1 & 0 \\
x_{a_1}^1 & x_{a_2}^1 & \cdots & x_{a_{k-2}}^1 & \rho^1 & \xi^1 \\
\vdots & \vdots & \vdots & \vdots & \vdots & \vdots \\
x_{a_1}^{k-1} & x_{a_2}^{k-1} & \cdots & x_{a_{k-2}}^{k-1} & \rho^{k-1} & \xi^{k-1}
\end{vmatrix} \\
&:= -2\Delta_{a_1 \cdots a_{k-2}\rho\,\xi}^{(k)} \,.
\end{aligned} \tag{5.7}
$$

The soft factor in Eq.(5.2) then becomes,

$$
\begin{aligned}
S_{\text{DS}}^{(k)} &= \int d^{k-1}\rho\, d^{k-1}\xi\, \delta^{(k-1)}(E_{n-1} + E_n)\, \delta^{(k-1)}(E_{n-1} - E_n) \\
&\quad \times \left[ \frac{|n-k \cdots 1||(n-k+1) \cdots 1\, 2| \cdots |n-2\, 1 \cdots k-1|}{\Delta_{(n-k+1) \cdots n-2\rho\,\xi}^{(k)} \cdots \Delta_{1 \cdots k-2\rho\,\xi}^{(k)} |n-k \cdots n-2\, \rho||\rho\, 1 \cdots k-1|} \right]^2 \,. \tag{5.8}
\end{aligned}
$$

The last two equations in (5.3) can be expressed as,

$$
E_{n-1}^{(i)} = \sum_{1 \le a_1 \cdots < a_{k-1} \le n-2} \frac{s_{a_1 \cdots a_{k-1}\, n-1}}{|a_1 \cdots a_{k-1}\, \rho|} \frac{\partial}{\partial \rho^i} |a_1 \cdots a_{k-1}\, \rho|
$$

$$+ \sum_{1 \le a_1 \cdots < a_{k-2} \le n-2} \frac{s_{a_1 \cdots a_{k-2}\, n-1\, n}}{2 \Delta^{(k)}_{a_1 \cdots a_{k-2}\, \rho\, \xi}} \frac{\partial}{\partial \xi^i} \Delta^{(k)}_{a_1 \cdots a_{k-2}\, \rho\, \xi} = 0,$$

$$E_n^{(i)} = \sum_{1 \le a_1 \cdots < a_{k-1} \le n-2} \frac{s_{a_1 \cdots a_{k-1}\, n}}{|a_1 \cdots a_{k-1}\, \rho|} \frac{\partial}{\partial \rho^i} |a_1 \cdots a_{k-1}\, \rho|$$

$$- \sum_{1 \le a_1 \cdots < a_{k-2} \le n-2} \frac{s_{a_1 \cdots a_{k-2}\, n-1\, n}}{2 \Delta^{(k)}_{a_1 \cdots a_{k-2}\, \rho\, \xi}} \frac{\partial}{\partial \xi^i} \Delta^{(k)}_{a_1 \cdots a_{k-2}\, \rho\, \xi} = 0. \tag{5.9}$$

We first do the $\xi$ integrals in Eq.(5.8), for which we make a change of variables,

$$\begin{aligned} \xi^1 &= \epsilon, \\ \xi^j &= \epsilon \zeta^j, \quad j = 2, \cdots k-1. \end{aligned} \tag{5.10}$$

This implies,

$$d^{k-1} \xi = \epsilon^{k-2} d\epsilon \, d^{k-2} \zeta. \tag{5.11}$$

Following the above change of variables we can write $\Delta^{(k)}_{a_1 \cdots a_{k-2}\, \rho\, \xi}$ as,

$$\Delta^{(k)}_{a_1 \cdots a_{k-2}\, \rho\, \xi} = \epsilon \left( x^1_{a_1} - \rho^1 \right) \cdots \left( x^1_{a_{k-2}} - \rho^1 \right) \Delta^{(k-1)}_{\zeta\, a_1 \cdots a_{k-2}}, \tag{5.12}$$

where we have defined,

$$\Delta^{(k-1)}_{\zeta\, a_1 \cdots a_{k-2}} := \begin{vmatrix} 1 & 1 & \cdots & 1 \\ \zeta^2 & \alpha^2_{a_1} & \cdots & \alpha^2_{a_{k-2}} \\ \vdots & \vdots & \vdots & \vdots \\ \zeta^{k-1} & \alpha^{k-1}_{a_1} & \cdots & \alpha^{k-1}_{a_{k-2}} \end{vmatrix}, \qquad \alpha^j_a = \frac{x^j_a - \rho^j}{x^1_a - \rho^1}. \tag{5.13}$$

Similarly $\Delta^{(k-1)}_{a_1 \cdots a_{k-2}\, a_{k-1}}$ will be denoted by the determinant of $(k-1) \times (k-1)$ minor obtained by replacing the column $\begin{pmatrix} 1 \\ \zeta^2 \\ \vdots \\ \zeta^{k-1} \end{pmatrix}$ by $\begin{pmatrix} 1 \\ \alpha^2_{k-1} \\ \vdots \\ \alpha^{k-1}_{k-1} \end{pmatrix}$ in Eq.(5.13).

We define a new potential function,

$$\tilde{S} = \sum_{1 \le a_1 \cdots < a_{k-2} \le n-2} s_{a_1 \cdots a_{k-2}\, n-1\, n} \log \Delta^{(k)}_{a_1 \cdots a_{k-2}\, \rho\, \xi}, \tag{5.14}$$

such that near $\epsilon \to 0$, $E^{(i)}_{n-1} - E^{(i)}_n$ can be approximated to be,

$$E^{(i)}_{n-1} - E^{(i)}_n \approx \sum_{1 \le a_1 \cdots < a_{k-2} \le n-2} \frac{s_{a_1 \cdots a_{k-2}\, n-1\, n}}{\Delta^{(k)}_{a_1 \cdots a_{k-2}\, \rho\, \xi}} \frac{\partial}{\partial \xi^i} \Delta^{(k)}_{a_1 \cdots a_{k-2}\, \rho\, \xi} = \frac{\partial}{\partial \xi^i} \tilde{S} = 0. \tag{5.15}$$

The delta function with the argument $E^{(i)}_{n-1} - E^{(i)}_n$ can be used to perform $\xi$ integration as a contour integral. Under the change of coordinates from $\xi^i$ to $(\epsilon, \zeta^j)$ this delta function transforms as,

$$\delta^{(k-1)} (E_{n-1} - E_n) = \epsilon^{k-2} \delta \left( \frac{\partial \tilde{S}}{\partial \epsilon} \right) \delta^{(k-2)} \left( \frac{\partial \tilde{S}}{\partial \zeta^j} \right). \tag{5.16}$$

$\epsilon$ integration gives a residue $\left( \sum_{1 \le a_1 \cdots < a_{k-2} \le n-2} s_{a_1 \cdots a_{k-2}\, n-1\, n} \right)^{-1}$. The soft factor in Eq.(5.8) now takes the form,

$$S^{(k)}_{\mathrm{DS}} = \frac{1}{\sum_{1 \le a_1 \cdots < a_{k-2} \le n-2} s_{a_1 \cdots a_{k-2}\, n-1\, n}} \int d^{k-2} \zeta \, \delta^{(k-2)} \left( \frac{\partial \tilde{S}}{\partial \zeta^j} \right)$$

$$\times \left[ \frac{\Delta^{(k-1)}_{(n-k+1)\cdots 1} \cdots \Delta^{(k-1)}_{n-2\,1\cdots k-2}}{\Delta^{(k-1)}_{(n-k+1)\cdots n-2\,\zeta} \cdots \Delta^{(k-1)}_{n-2\,\zeta\cdots 1} \Delta^{(k-1)}_{\zeta\,1\cdots k-2}} \right]^2 \int d^{k-1}\rho \; \delta^{(k-1)}(E_{n-1}+E_n)$$

$$\times \left[ \frac{|n-k\cdots 1||n-(k-1)\cdots 1\,2|\cdots|n-2\,1\cdots k-1|}{|n-k\cdots n-2\,\rho||(n-k+1)\cdots\rho\,1|\cdots|\rho\,1\cdots k-1|} \right]^2$$

$$= \frac{1}{\displaystyle\sum_{1\le a_1\cdots < a_{k-2}\le n-2} s_{a_1\cdots a_{k-2}\,n-1\,n}} \; \mathsf{S}^{(k-1)}\!\left(s_{a_1\cdots a_{k-2}\,m} \to s_{a_1\cdots a_{k-2}\,n-1\,n}\right) \mathsf{S}^{(k)}. \qquad (5.17)$$

This is the expression for double soft factor for arbitrary $k$ when adjacent particles are taken to be soft. $\mathsf{S}^{(k-1)}$ is the single soft factor corresponding to $k-1$ which has the pole structure of the form $s_{a_1\cdots a_{k-2}\,n-1\,n}$ in place of usual $(k-1)$-indexed Mandelstam variables. $\mathsf{S}^{(k)}$ denotes the single soft factor for $k$ with shifted propagators of the form $s_{a_1\cdots a_{k-1}\,n-1} + s_{a_1\cdots a_{k-1}\,n}$. Eq.(5.17) provides a recursion relation of calculating double soft factor from single soft factors. It is worth pointing out that the leading contribution to the double soft factor for arbitrary $k$ scales at $\tau^{-3(k-1)}$.

## 6  Next to Adjacent Soft Limits for $k=3$ Amplitudes

In this section, we study the double soft limits for next to adjacent external states labelled by $n-2$ and $n$. For this, we choose the scaling of the generalised Mandelstam variables as,

$$s_{a\,b\,n-2} = \tau \hat{s}_{a\,b\,n-2}, \qquad s_{a\,b\,n} = \tau \hat{s}_{a\,b\,n}, \qquad s_{a\,n-2\,n} = \tau^2 \hat{s}_{a\,n-2\,n}, \qquad a,b \ne n-2,n, \quad (6.1)$$

where, $a,b$ label the hard punctures. We will consider regular solutions and ignore the singular solutions where the determinants $|a\;b\;n|$, and $|a\;b\;n-2|$ scale as $\tau$, the scattering equations become,

$$E_a^{(i)} = \sum_{b,c\ne a,n-2,n} \frac{s_{a\,b\,c}}{|a\,b\,c|} \frac{\partial}{\partial x_a^{(i)}} |a\,b\,c| = 0, \quad \forall a,$$

$$E_{n-2}^{(i)} = \tau \sum_{a,b\ne n-2,n} \frac{\hat{s}_{a\,b\,n-2}}{|a\,b\,n-2|} \frac{\partial}{\partial x_{n-2}^{(i)}} |a\,b\,n-2|$$

$$+ \tau^2 \sum_{a\ne n-2,n} \frac{\hat{s}_{a\,n-2\,n}}{|a\,n-2\,n|} \frac{\partial}{\partial x_{n-2}^{(i)}} |a\,n-2\,n| = 0, \qquad (6.2)$$

$$E_n^{(i)} = \tau \sum_{a,b\ne n-2,n} \frac{\hat{s}_{a\,b\,n}}{|a\,b\,n|} \frac{\partial}{\partial x_n^{(i)}} |a\,b\,n|$$

$$+ \tau^2 \sum_{a\ne n-2,n} \frac{\hat{s}_{a\,n-2\,n}}{|a\,n-2\,n|} \frac{\partial}{\partial x_n^{(i)}} |a\,n-2\,n| = 0.$$

The soft factor for the next to adjacent soft punctures can then be expressed as,

$$\mathsf{S}^{(3)}_{n,n-2} = \int d^2 x_n \int d^2 x_{n-2} \; \delta^{(2)}(E_n)\delta^{(2)}(E_{n-2})$$

$$\times \left[ \frac{|n-4\,n-3\,n-1||n-3\,n-1\,1||n-1\,1\,2|}{|n-4\,n-3\,n-2||n-3\,n-2\,n-1||n-2\,n-1\,n||n-1\,n\,1||n\,1\,2|} \right]^2. \qquad (6.3)$$

In the next subsection, we will study the consecutive double soft limit to understand the leading order behaviour of the soft factor and later we will take up the simultaneous double soft limit as a consistency check on the results of the consecutive soft limit.

## 6.1 Consecutive double soft limit

In the consecutive double soft limit, we will encounter two possibilities depending on which external state becomes soft at a faster rate.

**a)**  We will start with the case where the $n$-th particle is softer than the $(n-2)$-th one where $s_{n\,a\,b} = \tau_1 \hat{s}_{n\,a\,b}$ and $s_{n-2\,a\,b} = \tau_2 \hat{s}_{n-2\,a\,b}$, with $\tau_1 \ll \tau_2$. After taking these two limits we have,

$$m_n^{(3)}(\text{I}|\text{I}) = \mathsf{S}_{n\to 0}^{(3)}\Big|_{n-2\to 0} \mathsf{S}_{n-2\to 0}^{(3)} m_{n-2}^{(3)}(\text{I}|\text{I}), \tag{6.4}$$

where,

$$\mathsf{S}_{n-2\to 0}^{(3)} = \frac{1}{\tau_2^2}\Bigg[\frac{1}{\hat{s}_{n-4\,n-3\,n-2}\,\hat{s}_{n-2\,n-1\,1}} + \frac{1}{\displaystyle\sum_{a=1}^{n-3}\hat{s}_{n-2\,a\,n-1}}\left(\frac{1}{\hat{s}_{n-3\,n-2\,n-1}} + \frac{1}{\hat{s}_{n-2\,n-1\,1}}\right)$$
$$+\frac{1}{\displaystyle\sum_{a=1,a\neq n-2,n-3}^{n-1}\hat{s}_{n-2\,a\,n-3}}\left(\frac{1}{\hat{s}_{n-3\,n-2\,n-1}} + \frac{1}{\hat{s}_{n-4\,n-3\,n-2}}\right)\Bigg], \tag{6.5}$$

and,

$$\mathsf{S}_{n\to 0}^{(3)}\Big|_{n-2\to 0} = \frac{1}{\tau_1^2}\Bigg\{\frac{1}{\tau_2\hat{s}_{n-2\,n-1\,n}\,\hat{s}_{n\,1\,2}} + \frac{1}{\displaystyle\sum_{a=2}^{n-1}\hat{s}_{1\,a\,n}}\left(\frac{1}{\hat{s}_{n-1\,n\,1}} + \frac{1}{\hat{s}_{n\,1\,2}}\right)$$
$$+\frac{1}{\displaystyle\sum_{a=1}^{n-2}\hat{s}_{a\,n-1\,n}}\left(\frac{1}{\tau_2\hat{s}_{n-2\,n-1\,n}} + \frac{1}{\hat{s}_{n-1\,n\,1}}\right)\Bigg\}$$
$$= \frac{1}{\tau_1^2\tau_2}\left\{\frac{1}{\hat{s}_{n-2\,n-1\,n}\,\hat{s}_{n\,1\,2}} + \frac{1}{\displaystyle\sum_{a=1}^{n-3}\hat{s}_{a\,n-1\,n}}\frac{1}{\hat{s}_{n-2\,n-1\,n}}\right\}. \tag{6.6}$$

The full soft factor for this consecutive limit with $\tau_1 \ll \tau_2$ is,

$$\mathsf{S}_{n\to 0}^{(3)}\Big|_{n-2\to 0}\mathsf{S}_{n-2\to 0}^{(3)} = \frac{1}{\tau_1^2\tau_2^3}\left\{\frac{1}{\hat{s}_{n-2\,n-1\,n}}\left(\frac{1}{\hat{s}_{n\,1\,2}} + \frac{1}{\displaystyle\sum_{a=1}^{n-3}\hat{s}_{a\,n-1\,n}}\right)\right\}\Bigg[\frac{1}{\hat{s}_{n-4\,n-3\,n-2}\,\hat{s}_{n-2\,n-1\,1}}$$
$$+\frac{1}{\displaystyle\sum_{a=1,a\neq n-2,n-1}^{n-3}\hat{s}_{n-2\,a\,n-1}}\left(\frac{1}{\hat{s}_{n-3\,n-2\,n-1}} + \frac{1}{\hat{s}_{n-2\,n-1\,1}}\right)$$
$$+\frac{1}{\displaystyle\sum_{a=1,a\neq n-2,n-3}^{n-1}\hat{s}_{n-2\,a\,n-3}}\left(\frac{1}{\hat{s}_{n-3\,n-2\,n-1}} + \frac{1}{\hat{s}_{n-4\,n-3\,n-2}}\right)\Bigg]. \tag{6.7}$$

**b)**  If instead we take $\tau_2 \ll \tau_1$ and carry out a similar analysis as in the previous case, the consecutive soft factor becomes,

$$\mathsf{S}_{n-2\to 0}^{(3)}\Big|_{n\to 0}\mathsf{S}_{n\to 0}^{(3)} = \frac{1}{\tau_1^3\tau_2^2}\left\{\frac{1}{\hat{s}_{n-2\,n-1\,n}}\left(\frac{1}{\hat{s}_{n-4\,n-3\,n-2}} + \frac{1}{\displaystyle\sum_{a=1}^{n-3}\hat{s}_{n-2\,a\,n-1}}\right)\right\}$$

$$\times \left[ \frac{1}{\hat{s}_{n-3\,n-1\,n}\,\hat{s}_{n\,1\,2}} + \frac{1}{\displaystyle\sum_{a=2}^{n-1} \hat{s}_{1\,a\,n}} \left( \frac{1}{\hat{s}_{n-1\,n\,1}} + \frac{1}{\hat{s}_{n\,1\,2}} \right) \right.$$

$$\left. + \frac{1}{\displaystyle\sum_{a=1}^{n-3} \hat{s}_{a\,n-1\,n}} \left( \frac{1}{\hat{s}_{n-3\,n-1\,n}} + \frac{1}{\hat{s}_{n-1\,n\,1}} \right) \right] . \tag{6.8}$$

It is now evident from Eq. (6.7) and Eq. (6.8) that the leading behaviour of the double soft factor in the simultaneous limit is expected to be $\mathcal{O}(\tau^{-5})$. On the other hand, the scaling of the soft factor for the non-degenerate solutions goes as $\mathcal{O}(\tau^{-4})$, which is subleading compared to the degenerate case. This suggests that if we want to pick up the leading effects in the simultaneous limit, it suffices to look at the degenerate solutions. However, in appendix A, we have analysed the contribution of non-degenerate solutions to the subleading results.

## 6.2 Simultaneous double soft limit: Degenerate solutions

We will now look at the simultaneous double soft limit. In this case the determinant $|a\,n-2\,n|$ scales as $\tau$, for $a$ belonging to hard particles. This scaling happens when either the two soft punctures come close to each other, or they are nearly collinear with one of the hard punctures. We will look at both these possibilities now.

### 6.2.1 Collision configuration

To study the colliding configurations we consider the following change of variables,

$$x_{n-2}^i = \rho^i + \xi^i, \qquad x_n^i = \rho^i - \xi^i, \qquad i = 1, 2 , \tag{6.9}$$

where $\xi$ is $\mathcal{O}(\tau)$. In terms of these new variables, scattering equations become,

$$E_{n-2}^{(i)} + E_n^{(i)} = \sum_{a,b} \frac{s_{a\,b\,n-2} + s_{a\,b\,n}}{|a\,b\,\rho|} \frac{\partial}{\partial \rho^i} |a\,b\,\rho| ,$$

$$E_{n-2}^{(i)} - E_n^{(i)} = \sum_{a,b} \frac{s_{a\,b\,n-2} - s_{a\,b\,n}}{|a\,b\,\rho|} \frac{\partial}{\partial \rho^i} |a\,b\,\rho| + \sum_{a} \frac{s_{a\,n-2\,n}}{|a\,\rho\,\xi|} \frac{\partial}{\partial \xi^i} |a\,\rho\,\xi| . \tag{6.10}$$

The soft factor for the degenerate solutions can be expressed as,

$$\mathsf{S}_{\text{deg}}^{(3)} = \int d^2\rho \; d^2\xi \; \delta^{(2)}(E_{n-2} + E_n) \delta^{(2)}(E_{n-2} - E_n)$$

$$\times \left[ \frac{|n-4\,n-3\,n-1||n-3\,n-1\,1||n-1\,1\,2|}{|n-4\,n-3\,\rho||n-3\,\rho\,n-1||\rho\,n-1\,\xi||n-1\,\rho\,1||\rho\,1\,2|} \right]^2 . \tag{6.11}$$

We can first do the $\xi$ integration by deforming the contour. To do that we will consider following parametrisation,

$$\xi^1 = \epsilon ,$$
$$\xi^2 = \epsilon \alpha . \tag{6.12}$$

The integration measure can be written as,

$$d^2\xi = \epsilon \, d\epsilon \, d\alpha . \tag{6.13}$$

Similarly delta functions will transform with a Jacobian factor, which is equal to $\epsilon$. We can then express the $\xi$ integral in terms of $\epsilon$ and $\alpha$. It is easy to see that there is no singularity in the $\epsilon$ contour integral and as a result this integral vanishes,

$$\oint \frac{\epsilon^2 d\epsilon \, d\alpha}{\frac{1}{\epsilon}\sum_a s_{a\,n-2\,n}\sum_a \frac{s_{a\,n-2\,n}}{\alpha-\alpha_a}}\left[\frac{1}{\epsilon(\rho^1-x^1_{n-1})(\alpha-\alpha_{n-1})}\right]^2, \qquad \alpha_a = \frac{x_a^2-\rho^2}{x_a^1-\rho^1}$$

$$=\frac{1}{\sum_a s_{a\,n-2\,n}}\oint \epsilon \, d\epsilon \oint \frac{d\alpha}{\sum_a \frac{s_{a\,n-2\,n}}{\alpha-\alpha_a}}\left[\frac{1}{(\rho^1-x^1_{n-1})(\alpha-\alpha_{n-1})}\right]^2$$

$$=0\,. \tag{6.14}$$

In fact, the collision singularity contributes at the order $\tau^{-4}$, and in that sense, this vanishing contribution is subleading. This hints that the dominating contribution comes from the collinear singularity, where two soft punctures become collinear with one hard puncture.

### 6.2.2 Soft punctures collinear to one hard puncture

We will now consider the case when the determinant $|n-2\,n\,d|$ scales as $\tau$, *i.e.*, both the soft punctures are nearly collinear to one of the hard punctures labelled by $d$. We will now use the collinearity ansatz by parametrising $x^i_{n-2}$ as,

$$x_{n-2} = x_d + \alpha(x_n - x_d)\,,$$
$$y_{n-2} = y_d + \alpha(y_n - y_d) + \tau\xi\,. \tag{6.15}$$

Therefore the determinant $|n-2\,n\,d|$ becomes,

$$|n-2\,n\,d| = \tau\xi(x_d - x_n)\,. \tag{6.16}$$

The Parke-Taylor contribution for $d = n-1$ then can be written as,

$$\left[\frac{|n-4\,n-3\,n-1||n-3\,n-1\,1||n-1\,1\,2|}{\tau\xi(x_{n-1}-x_n)|n-4\,n-3\,\alpha(\sigma_n-\sigma_{n-1})||n-3\,\alpha(\sigma_n-\sigma_{n-1})\,n-1||n-1\,n\,1||n\,1\,2|}\right]^2, \tag{6.17}$$

where we have used the definition,

$$|a\,b\,\alpha(\sigma_n-\sigma_{n-1})| = |a\,b\,d| + \begin{vmatrix} 1 & 1 & 0 \\ x_a & x_b & \alpha(x_n-x_{n-1}) \\ y_a & y_b & \alpha(y_n-y_{n-1}) \end{vmatrix}. \tag{6.18}$$

For $d \neq n-1$, the Parke-Taylor contribution is $\mathcal{O}(\tau^0)$. For $d = n-1$, the measure becomes,

$$dx_n dy_n dx_{n-2} dy_{n-2} = \tau(x_n - x_{n-1}) dx_n dy_n d\alpha d\xi\,. \tag{6.19}$$

The scattering equations for $n$ and $n-2$ become,

$$E^{(1)}_{n-2} = \tau \sum_{1\le a<b\le n-1, a,b\neq n-2} \frac{\hat{s}_{a\,b\,n-2}}{|a\,b\,\alpha(\sigma_n-\sigma_{n-1})|}(y_a-y_b) - \tau\frac{\hat{s}_{n-2\,n-1\,n}}{\xi(x_n-x_{n-1})}(y_n-y_{n-1}),$$

$$E^{(2)}_{n-2} = -\tau \sum_{1\le a<b\le n-1, a,b\neq n-2} \frac{\hat{s}_{a\,b\,n-2}}{|a\,b\,\alpha(\sigma_n-\sigma_{n-1})|}(x_a-x_b) + \tau\frac{\hat{s}_{n-2\,n-1\,n}}{\xi},$$

$$E_n^{(1)} = \tau \sum_{1 \le a < b \le n-1, a,b \ne n-2} \frac{\hat{s}_{a\,b\,n}}{|a\,b\,n|}(y_a - y_b) + \alpha \tau \frac{\hat{s}_{n-2\,n-1\,n}}{\xi(x_n - x_{n-1})}(y_n - y_{n-1}),$$

$$E_n^{(2)} = -\tau \sum_{1 \le a < b \le n-1, a,b \ne n-2} \frac{\hat{s}_{a\,b\,n}}{|a\,b\,n|}(x_a - x_b) - \tau \alpha \frac{\hat{s}_{n-2\,n-1\,n}}{\xi}. \tag{6.20}$$

Thus each scattering equation is of $\mathcal{O}(\tau)$. Therefore in the soft factor, we have one power of $\tau$ in the numerator from the measure, four powers of $\tau$ in the denominator from the scattering equations and two powers of $\tau$ in denominator from the Parke-Taylor factor. Thus, this limit contributes at order $\tau^{-5}$ and hence it is the leading order contribution for the next to adjacent puncture soft limit. However, we have not found suitable linear combinations of these equations that would allow us to independently deform the contours away from the scattering equations. On the other hand, if we localise the delta function on variables $\xi$ and $\alpha$, while we can consider any of the scattering equations above and solve for $\xi$, we find that solving for $\alpha$ leads to an $n-3$ degree polynomial equation. Thus for generic parametrisation, the computation seems to be analytically intractable. We hope to return to this in the future.

## 7 Next to Next to Adjacent Soft Limits for $k = 3$ Amplitudes

In this section we will study the double soft limit for $n$-th puncture and its next to next to adjacent puncture. We consider punctures labelled by $n-3$ and $n$ to be soft,

$$s_{a\,b\,n-3} = \tau \hat{s}_{a\,b\,n-3}, \qquad s_{a\,b\,n} = \tau \hat{s}_{a\,b\,n}, \qquad s_{a\,n-3\,n} = \tau^2 \hat{s}_{a\,n-3\,n}, \qquad a, b \ne n-3, n. \tag{7.1}$$

The scattering equations for regular solutions, *i.e.*, when the determinants $|a\,b\,n|$ and $|a\,b\,n-3|$ are non-vanishing as $\tau \to 0$, are,

$$E_a^{(i)} = \sum_{b,c \ne a,n-3,n} \frac{s_{a\,b\,c}}{|a\,b\,c|} \frac{\partial}{\partial x_a^{(i)}} |a\,b\,c| = 0, \quad \forall a,$$

$$E_{n-3}^{(i)} = \tau \sum_{a,b \ne n-3,n} \frac{\hat{s}_{a\,b\,n-3}}{|a\,b\,n-3|} \frac{\partial}{\partial x_{n-3}^{(i)}} |a\,b\,n-3|$$
$$+ \tau^2 \sum_{a \ne n-3,n} \frac{\hat{s}_{a\,n-3\,n}}{|a\,n-3\,n|} \frac{\partial}{\partial x_{n-3}^{(i)}} |a\,n-3\,n| = 0,$$

$$E_n^{(i)} = \tau \sum_{a,b \ne n-3,n} \frac{\hat{s}_{a\,b\,n}}{|a\,b\,n|} \frac{\partial}{\partial x_n^{(i)}} |a\,b\,n|$$
$$+ \tau^2 \sum_{a \ne n-3,n} \frac{\hat{s}_{a\,n-3\,n}}{|a\,n-3\,n|} \frac{\partial}{\partial x_n^{(i)}} |a\,n-3\,n| = 0. \tag{7.2}$$

Hence the soft factor can be written as,

$$S_{n,n-3}^{(3)} = \int d^2 x_n \int d^2 x_{n-3} \, \delta^{(2)}(E_n) \delta^{(2)}(E_{n-3}) \Bigg[ \frac{|n-5\,n-4\,n-2|}{|n-5\,n-4\,n-3||n-4\,n-3\,n-2|}$$
$$\times \frac{|n-4\,n-2\,n-1||n-2\,n-1\,1||n-1\,1\,2|}{|n-3\,n-2\,n-1||n-2\,n-1\,n||n-1\,n\,1||n\,1\,2|} \Bigg]^2. \tag{7.3}$$

In this case contributions from non-degenerate solutions dominate over that of degenerate ones because in the Parke-Taylor factor none of the determinants simultaneously contain the punctures $n$ and $(n-3)$, hence it does not lead to any $\mathcal{O}(\tau)$ contribution in the denominator for the degenerate case. As a result in the degenerate case, the measure and the scattering

equations together give the scaling which is less than the $\mathcal{O}(\tau^{-4})$ scaling of the non-degenerate solutions. In this case the soft factor is expressed as,

$$
\begin{aligned}
\mathrm{S}_{n,n-3}^{(3)} &= \int d^2 x_n\, \delta^{(2)}(E_n)\left[\frac{|n-2\,n-1\,1||n-1\,1\,2|}{|n-2\,n-1\,n||n-1\,n\,1||n\,1\,2|}\right]^2 \\
&\times \int d^2 x_{n-3}\,\delta^{(2)}(E_{n-3})\left[\frac{|n-5\,n-4\,n-2||n-4\,n-2\,n-1|}{|n-5\,n-4\,n-3||n-4\,n-3\,n-2||n-3\,n-2\,n-1|}\right]^2
\end{aligned}
$$

$$
\begin{aligned}
=\frac{1}{\tau^4}&\left\{\frac{1}{\hat{s}_{n-2\,n-1\,n}\,\hat{s}_{n\,1\,2}} + \frac{1}{\displaystyle\sum_{\substack{a=2 \\ a\neq n-3}}^{n-1}\hat{s}_{a\,n\,1}}\left(\frac{1}{\hat{s}_{n-1\,n\,1}} + \frac{1}{\hat{s}_{n\,1\,2}}\right)\right. \\
&\left.+\frac{1}{\displaystyle\sum_{\substack{a=1 \\ a\neq n-3}}^{n-2}\hat{s}_{a\,n-1\,n}}\left(\frac{1}{\hat{s}_{n-1\,n\,1}} + \frac{1}{\hat{s}_{n-2\,n-1\,n}}\right)\right\}\left[\frac{1}{\hat{s}_{n-5\,n-4\,n-3}\,\hat{s}_{n-3\,n-2\,n-1}}\right.\\
&+\frac{1}{\displaystyle\sum_{\substack{a=1 \\ a\neq n-3,\,n-4}}^{n-1}\hat{s}_{a\,n-4\,n-3}}\left(\frac{1}{\hat{s}_{n-5\,n-4\,n-3}} + \frac{1}{\hat{s}_{n-4\,n-3\,n-2}}\right)\\
&\left.+\frac{1}{\displaystyle\sum_{\substack{a=1 \\ a\neq n-3,\,n-2}}^{n-1}\hat{s}_{a\,n-3\,n-2}}\left(\frac{1}{\hat{s}_{n-4\,n-3\,n-2}} + \frac{1}{\hat{s}_{n-3\,n-2\,n-1}}\right)\right]
\end{aligned}
$$

$$
= \mathrm{S}_n^{(3)}\mathrm{S}_{n-3}^{(3)}\,, \tag{7.4}
$$

where $\mathrm{S}_n^{(3)}$ and $\mathrm{S}_{n-3}^{(3)}$ are the single soft factors corresponding to $n$-th and $(n-3)$-th soft external states respectively. Similar result holds when any of the external state from the set $\{3,4,\cdots n-3\}$ along with $n$-th state are taken to be soft,

$$
\mathrm{S}_{n,a}^{(3)} = \mathrm{S}_n^{(3)}\mathrm{S}_a^{(3)} \qquad \forall\, a \in \{3,4,\cdots n-3\}\,. \tag{7.5}
$$

Therefore, in this case double soft factor is the same as the product of two single soft factors. This is analogous to the $k=2$ non-adjacent case discussed in 2.2.2.

# 8 Discussion

In this paper, we derived the double soft limit for adjacent soft external states for arbitrary $k$. We also generalised our method to the double soft limit of the next to adjacent and next to next to adjacent soft external states. We found that in the simultaneous double soft limit leading contribution in case of the adjacent soft external states scales as $\tau^{-3(k-1)}$ in the $\tau \to 0$ limit. It follows from a simple scaling argument, in the canonical color ordering, that when the soft labels $i$ and $j$ in an amplitude are arranged in such a way that $|i-j| \in \{k, k+1, \cdots n-k+1\}$ then the double soft factor is a product of two single soft factors and hence it scales as $\tau^{-2(k-1)}$. For all intermediate separations, i.e., $1 < |i-j| < k$, the scaling exponent is linear in $|i-j|$ for a cyclic ordering.

The fact that these soft limits involve higher order poles seems to indicate that these am-

plitudes could be relevant for computation of loop diagrams, see e.g., [45][4]. Alternatively, the higher order poles could be a signature of composite particles or multiparticle states contributing to the amplitude. It would be interesting to check if this is true by further studying the factorisation in detail. In a recent paper by some of the authors [69], the relation of the double soft factor with the cluster algebra was presented. It would be interesting to generalise our results to multiple soft theorem. There are no efficient techniques to date that can compute arbitrary $(k, n)$ amplitudes. However, one may be able to bootstrap these amplitudes by knowing the structure of multi-soft factors. Whether multi-soft factors themselves are amplitudes of some theory is worth exploring but we will leave it for the future.

## Acknowledgements

We would like to thank Subhroneel Chakrabarti, Alok Laddha, R. Loganayagam, Biswajit Sahoo, and Ashoke Sen for discussion. One of us (APS) would like to thank ICTS for virtual hospitality and the participants of the seminar for asking interesting and pertinent questions. We would like to thank the organisers and the participants of the Recent Developments in S-matrix Theory program (ICTS/rdst2020/07) where a preliminary version of this work was presented.

## A  Next to Adjacent Non-degenerate Solutions for $k = 3$

In this appendix, we will discuss the non-degenerate contributions to the subleading results for the next to adjacent double soft limit. For the non-degenerate configuration we neglect terms of $\mathcal{O}(\tau^2)$ in the scattering equations Eq. (6.2). In the homogeneous coordinates we then obtain,

$$
\begin{aligned}
\text{S}^{(3)}_{\text{non-deg}} = \oint \frac{\left(\sigma_n\, d^2\sigma_n\right)(X_1\, Y_1\, \sigma_n)}{\sum\limits_{b,c} \frac{s_{n\,b\,c}(X_1\,b\,c)}{(\sigma_n\,b\,c)} \sum\limits_{b,c} \frac{s_{n\,b\,c}(Y_1\,b\,c)}{(\sigma_n\,b\,c)}} \oint \frac{\left(\sigma_{n-2}\, d^2\sigma_{n-2}\right)(X_2\, Y_2\, \sigma_{n-2})}{\sum\limits_{b,c} \frac{s_{n-2\,b\,c}(X_2\,b\,c)}{(\sigma_{n-2}\,b\,c)} \sum\limits_{b,c} \frac{s_{n-2\,b\,c}(Y_2\,b\,c)}{(\sigma_{n-2}\,b\,c)}} \\
\times \left[ \frac{(n-4\,n-3\,n-1)(n-3\,n-1\,1)(n-1\,1\,2)}{(n-4\,n-3\,\sigma_{n-2})(n-3\,\sigma_{n-2}\,n-1)(\sigma_{n-2}\,n-1\,\sigma_n)(n-1\,\sigma_n\,1)(\sigma_n\,1\,2)} \right]^2 .
\end{aligned} \tag{A.1}
$$

To evaluate this integral we will employ the global residue theorem, for which we will deform the contour from the poles of the scattering equations and pick up contributions from the singularities of the integrand. First we deform the $\sigma_{n-2}$ contour and encounter poles at the zeroes of the determinants in the denominator of the integrand,

$$
(n-4\,n-3\,\sigma_{n-2})\,, \qquad (n-3\,\sigma_{n-2}\,n-1)\,, \qquad (\sigma_{n-2}\,n-1\,\sigma_n)\,. \tag{A.2}
$$

While evaluating the integral in Eq. (A.1), we encounter two different kinds of singularities which we will discuss below by considering those cases one at a time.

### A.1  Collision singularities

In this case $(n-2)$-th puncture collides with any one of the punctures listed below,

$$
1. \qquad \sigma_{n-2} \;\rightarrow\; \sigma_{n-3}
$$

---

[4]Relation between the $(3, 6)$ amplitude and four point one-loop integrand in cubic biadjoint scalar field theory is studied in [69].

$$
\begin{aligned}
2. && \sigma_{n-2} &\rightarrow \sigma_{n-1} \\
3. && \sigma_{n-2} &\rightarrow \sigma_{n-4} \\
4. && \sigma_{n-2} &\rightarrow \sigma_{n}
\end{aligned} \tag{A.3}
$$

**Case 1: $\sigma_{n-2} \rightarrow \sigma_{n-3}$**

We choose the following parametrisation,

$$
\sigma_{n-2} = \sigma_{n-3} + \epsilon A, \tag{A.4}
$$

where $\epsilon$ is treated as one complex variable, which implies that the variable $A \in \mathbb{CP}^2$ has only one independent component. We choose $X_2 = \sigma_{n-3}$, and deform the contour of $\sigma_{n-2}$,

$$
\begin{aligned}
&\oint \frac{(n-3\, A\, dA)\, \epsilon^2 d\epsilon\, (n-3\, Y_2\, A)}{\displaystyle\sum_{b,c\neq n-3} s_{n-2\, b\, c} \sum_{b\neq n-3} \frac{s_{n-2\, n-3\, b}(Y_2\, b\, n-3)}{\epsilon(A\, b\, n-3)}} \\
&\qquad\times \left[\frac{(n-4\, n-3\, n-1)}{\epsilon^2\,(n-4\, n-3\, A)(n-3\, A\, n-1)(n-3\, n-1\, \sigma_n)}\right]^2 \\
=\ &\frac{1}{\displaystyle\sum_{b,c\neq n-3} s_{n-2\, b\, c}} \oint \frac{(n-3\, A\, dA)(n-3\, Y_2\, A)}{\displaystyle\sum_{b\neq n-3} \frac{s_{n-2\, n-3\, b}(Y_2\, b\, n-3)}{(A\, b\, n-3)}} \\
&\qquad\times \left[\frac{(n-4\, n-3\, n-1)}{(n-4\, n-3\, A)(n-3\, A\, n-1)}\right]^2 \frac{1}{(n-3\, n-1\, \sigma_n)^2} \\
=\ &\frac{1}{\displaystyle\sum_{c\neq n-3} s_{n-2\, n-3\, c}} \left[\frac{1}{s_{n-2\, n-3\, n-4}} + \frac{1}{s_{n-1\, n-2\, n-3}}\right] \frac{1}{(n-3\, n-1\, \sigma_n)^2}\ .
\end{aligned} \tag{A.5}
$$

In the last line we reduce the integration to that on $\mathbb{CP}^1$ by treating $n-3$ as a spectator, choose $Y_2 = \begin{pmatrix} 0 \\ 1 \end{pmatrix}$, and $A = \begin{pmatrix} 1 \\ x_A \end{pmatrix}$. We are then left with a $\sigma_n$ integration to be performed, however, we notice that it is precisely the single soft integral,

$$
\begin{aligned}
S_1^{(3)} =\ &\frac{1}{\displaystyle\sum_{c\neq n-3} s_{n-2\, n-3\, c}} \left[\frac{1}{s_{n-2\, n-3\, n-4}} + \frac{1}{s_{n-1\, n-2\, n-3}}\right] \\
&\qquad \times \oint \frac{(\sigma_n d^2\sigma_n)(X_1 Y_1 \sigma_n)}{\displaystyle\sum_{b,c}\frac{s_{nbc}(X_1 bc)}{(\sigma_n bc)} \sum_{b,c}\frac{s_{nbc}(Y_1 bc)}{(\sigma_n bc)}} \left[\frac{(n-3\, n-1\, 1)(n-1\, 1\, 2)}{(n-3\, n-1\, \sigma_n)(n-1\, \sigma_n\, 1)(\sigma_n 12)}\right]^2 \\
=\ &\left[\frac{1}{\displaystyle\sum_{c\neq n-3} s_{n-2\, n-3\, c}}\left(\frac{1}{s_{n-2\, n-3\, n-4}} + \frac{1}{s_{n-1\, n-2\, n-3}}\right)\right] \\
&\qquad \times \left[\frac{1}{s_{n-3\, n-1\, n}\, s_{n\, 1\, 2}} + \frac{1}{\displaystyle\sum_{c\neq n-1} s_{n-1\, n\, c}}\left(\frac{1}{s_{n-3\, n-1\, n}} + \frac{1}{s_{n-1\, n\, 1}}\right)\right. \\
&\qquad\qquad\qquad \left. + \frac{1}{\displaystyle\sum_{c\neq 1} s_{n\, 1\, c}}\left(\frac{1}{s_{n\, 1\, 2}} + \frac{1}{s_{n-1\, n\, 1}}\right)\right]. 
\end{aligned} \tag{A.6}
$$

**Case 2:** $\sigma_{n-2} \to \sigma_{n-1}$

We parametrize $\sigma_{n-2}$ as,

$$\sigma_{n-2} = \sigma_{n-1} + \epsilon A. \tag{A.7}$$

We choose the reference vector $X_2 = \sigma_{n-1}$. The $\sigma_{n-2}$ integration then becomes,

$$
\frac{1}{\sum\limits_{b,c \neq n-1} s_{n-2\,b\,c}} \oint \frac{(n-1\,A\,dA)(n-1\,Y_2\,A)}{\sum\limits_{c \neq n-1} \frac{s_{n-2\,n-1\,c}(Y_2\,n-1\,c)}{(A\,n-1\,c)}} \left[ \frac{(n-3\,n-1\,1)}{(n-3\,A\,n-1)(A\,n-1\,\sigma_n)} \right]^2
$$

$$
:= \frac{1}{\sum\limits_{b,c \neq n-1} s_{n-2\,b\,c}} \mathcal{F}(\sigma_n). \tag{A.8}
$$

The $\sigma_n$ integrations will then be as follows,

$$
S_2^{(3)} = \frac{1}{\sum\limits_{b,c \neq n-1} s_{n-2\,b\,c}} \oint \frac{(\sigma_n\,d^2\sigma_n)(X_1\,Y_1\,\sigma_n)}{\sum\limits_{b,c} \frac{s_{n\,b\,c}(X_1\,b\,c)}{(\sigma_n\,b\,c)} \sum\limits_{b,c} \frac{s_{n\,b\,c}(Y_1\,b\,c)}{(\sigma_n\,b\,c)}} \mathcal{F}(\sigma_n)
$$

$$
\times \left[ \frac{(n-1\,1\,2)}{(n-1\,\sigma_n\,1)(\sigma_n\,1\,2)} \right]^2. \tag{A.9}
$$

After contour deformation $\sigma_n$ will encounter poles at,

$$
\begin{array}{rrcl}
i. & \sigma_n & \to & \sigma_1 \\
ii. & \sigma_n & \to & \sigma_{n-1} \\
iii. & \sigma_n & \to & \sigma_2 \\
iv. & \sigma_n & \to & \text{Poles of } \mathcal{F}(\sigma_n)
\end{array} \tag{A.10}
$$

*i.* $\sigma_n \to \sigma_1$: We will choose,

$$\sigma_n = \sigma_1 + \eta B. \tag{A.11}$$

We set $X_1 = \sigma_1$ to obtain,

$$
S_{2;1}^{(3)} = \frac{1}{\sum\limits_{b,c \neq n-1} s_{n-2\,b\,c} \sum\limits_{e,f \neq 1} s_{n\,e\,f}} \oint \frac{(n-1\,A\,dA)(n-1\,Y_2\,A)}{\sum\limits_{c \neq n-1} \frac{s_{n-2\,n-1\,c}(Y_2\,n-1\,c)}{(A\,n-1\,c)}}
$$

$$
\times \left[ \frac{(n-3\,n-1\,1)}{(n-3\,A\,n-1)(A\,n-1\,1)} \right]^2 \oint \frac{(1\,B\,dB)(1\,Y_1\,B)}{\sum\limits_{c \neq 1} \frac{s_{n\,1\,c}(Y_1\,1\,c)}{B\,1\,c}} \left[ \frac{(n-1\,1\,2)}{(n-1\,B\,1)(B\,1\,2)} \right]^2
$$

$$
= \frac{1}{\sum\limits_{b \neq n-1} s_{n-2\,n-1\,b} \sum\limits_{c \neq 1} s_{n\,1\,c}} \left[ \frac{1}{s_{n-3\,n-2\,n-1}} + \frac{1}{s_{n-2\,n-1\,1}} \right] \left[ \frac{1}{s_{n-1\,n\,1}} + \frac{1}{s_{n\,1\,2}} \right]. \tag{A.12}
$$

*ii.* $\sigma_n \to \sigma_{n-1}$: We choose our parametrisation,

$$\sigma_n = \sigma_{n-1} + \eta B, \tag{A.13}$$

with the reference vector $X_2 = \sigma_{n-1}$. In this case we note that,

$$\mathcal{F}(\sigma_n) = \frac{1}{\eta^2} \mathcal{F}(B). \tag{A.14}$$

The $B$ integration then takes the form,

$$S_{2;2}^{(3)} = \frac{1}{\sum\limits_{b,c\neq n-1} s_{n-2\,b\,c} \sum\limits_{e,f\neq n-1} s_{n\,e\,f}}$$

$$\times \oint \frac{(n-1\,B\,dB)(n-1\,Y_2\,B)}{\sum\limits_{c\neq n-1} \frac{s_{n\,n-1\,c}(Y_1\,n-1\,1)}{(B\,n-1\,c)}} \left\{ \mathcal{F}(B) \frac{1}{(n-1\,B\,1)^2} \right\}. \qquad (A.15)$$

Here $\sigma_{n-1}$ is a spectator and hence both $A$ and $B$ integrals can be converted to $\mathbb{CP}^1$ integrals. We choose $Y_1 = Y_2 = \begin{pmatrix} 0 \\ 1 \end{pmatrix}$, $A = \begin{pmatrix} 1 \\ x_A \end{pmatrix}$, and $B = \begin{pmatrix} 1 \\ x_B \end{pmatrix}$. With this choice the above integrations become,

$$S_{2;2}^{(3)} = \frac{1}{\sum\limits_{b,c\neq n-1} s_{n-2\,b\,c} \sum\limits_{e,f\neq n-1} s_{n\,e\,f}} \oint \frac{dx_B}{\sum\limits_{c\neq n-1} \frac{s_{n\,n-1\,c}}{x_B-x_c}} \oint \frac{dx_A}{\sum\limits_{c\neq n-1} \frac{s_{n-2\,n-1\,c}}{x_A-x_c}}$$

$$\times \left[ \frac{x_{n-3}-x_1}{(x_A-x_{n-3})(x_A-x_B)(x_B-x_1)} \right]^2. \qquad (A.16)$$

We first evaluate residues at $x_A = x_{n-3}, x_B$ and then perform the $x_B$ integration. $x_A$ has a simple pole at $x_{n-3}$ and a double pole at $x_B$. Therefore we get,

$$S_{2;2}^{(3)} = \frac{1}{\sum\limits_{b,c\neq n-1} s_{n-2\,b\,c} \sum\limits_{e,f\neq n-1} s_{n\,e\,f}} \oint \frac{dx_B}{\sum\limits_{c\neq n-1} \frac{s_{n\,n-1\,c}}{x_B-x_c}} \left[ \frac{x_1-x_{n-3}}{(x_B-x_{n-3})(x_B-x_1)} \right]^2$$

$$\times \left[ \frac{1}{s_{n-1\,n-2\,n-3}} + \frac{\sum\limits_{c\neq n-1} \frac{s_{n-2\,n-1\,c}}{(x_B-x_c)^2}}{\left(\sum\limits_{c\neq n-1} \frac{s_{n-2\,n-1\,c}}{x_B-x_c}\right)^2} - \frac{2}{(x_B-x_{n-3})\sum\limits_{c\neq n-1} \frac{s_{n-2\,n-1\,c}}{x_B-x_c}} \right]. \qquad (A.17)$$

Thus we see $x_b$ has simple poles at $x_{n-3}$ and $x_1$. Moreover, the residue at $x_B = x_{n-3}$ turns out to vanish. The last term in the square bracket does not have a pole at $x_B = x_1$, hence evaluating $x_B$ integral we obtain,

$$S_{2;2}^{(3)} = \frac{1}{\sum\limits_{b\neq n-1} s_{n-2\,n-1\,b} \sum\limits_{c\neq n-1} s_{n-1\,n\,c}} \left[ \frac{1}{s_{n-1\,n\,1}} \left( \frac{1}{s_{n-3\,n-2\,n-1}} + \frac{1}{s_{n-2\,n-1\,1}} \right) \right]. \qquad (A.18)$$

*iii.* $\sigma_n \to \sigma_2$: We will now parametrize $\sigma_n$ as,

$$\sigma_n = \sigma_2 + \eta B. \qquad (A.19)$$

We find there is no pole as $\eta \to 0$. Hence there is no contribution from this singularity.

*iv.* **Poles of $\mathcal{F}(\sigma_n)$**: By definition of $\mathcal{F}(\sigma_n)$ in Eq.(A.8), and after performing the $A$ integration we obtain,

$$\mathcal{F}(\sigma_n) = \oint \frac{(n-1\,A\,dA)(n-1\,Y_2\,A)}{\sum\limits_{c\neq n-1} \frac{s_{n-2\,n-1\,c}(Y_2\,n-1\,c)}{(A\,n-1\,c)}} \left[ \frac{(n-3\,n-1\,1)}{(n-3\,A\,n-1)(A\,n-1\,\sigma_n)} \right]^2$$

$$= -\left[\frac{(n-3\,n-1\,1)}{(n-3\,n-1\,n)}\right]^2 \left[\frac{1}{s_{n-3\,n-2\,n-1}} + \frac{\sum_{c\neq n-1}\frac{s_{n-2\,n-1\,c}}{(n\,n-1\,c)^2}}{\left(\sum_{c\neq n-1}\frac{s_{n-2\,n-1\,c}}{(n\,n-1\,c)}\right)^2}\right.$$
$$\left. + \frac{2}{(n-3\,n-1\,n)\sum_{c\neq n-1}\frac{s_{n-2\,n-1\,c}}{(n\,n-1\,c)}}\right]. \quad (A.20)$$

From the above expression we can see that other than at $\sigma_n \to \sigma_{n-1}$ and at $\sigma_n \to \sigma_{n-3}$, which have already been taken into account in Eq.(A.18), no other poles of $\sigma_n$ appear from $\mathcal{F}(\sigma_n)$.

**Case 3: $\sigma_{n-2} \to \sigma_{n-4}$**

From the symmetry argument $\sigma_{n-2} \to \sigma_{n-4}$ does not contribute.

**Case 4: $\sigma_{n-2} \to \sigma_n$**

We choose our parametrization,

$$\sigma_{n-2} = \sigma_n + \epsilon A. \quad (A.21)$$

We will choose $X_2 = \sigma_n$ and obtain the Eq.(A.1) to be,

$$S_3^{(3)} = \oint \frac{\left(\sigma_n\,d^2\sigma_n\right)(X_1\,Y_1\,\sigma_n)}{\sum_{b,c}\frac{s_{n\,b\,c}(X_1\,b\,c)}{(\sigma_n\,b\,c)}\sum_{b,c}\frac{s_{n\,b\,c}(Y_1\,b\,c)}{(\sigma_n\,b\,c)}}\oint\frac{\left(\sigma_n\,A\,dA\right)\epsilon^2 d\epsilon\,(\sigma_n\,Y_2\,A)}{\sum_{b,c}\frac{s_{n-2\,b\,c}(\sigma_n\,b\,c)}{(\sigma_n+\epsilon A\,b\,c)}\sum_{b,c}\frac{s_{n-2\,b\,c}(Y_2\,b\,c)}{(\sigma_n+\epsilon A\,b\,c)}}$$
$$\times\left[\frac{(n-4\,n-3\,n-1)(n-3\,n-1\,1)(n-1\,1\,2)}{(n-4\,n-3\,\sigma_n)(n-3\,\sigma_n\,n-1)\epsilon(A\,n-1\,\sigma_n)(n-1\,\sigma_n\,1)(\sigma_n\,1\,2)}\right]^2$$
$$= \oint \frac{\left(\sigma_n\,d^2\sigma_n\right)(X_1\,Y_1\,\sigma_n)}{\sum_{b,c}\frac{s_{n\,b\,c}(X_1\,b\,c)}{(\sigma_n\,b\,c)}\sum_{b,c}\frac{s_{n\,b\,c}(Y_1\,b\,c)}{(\sigma_n\,b\,c)}}$$
$$\times\oint\frac{\left(\sigma_n\,A\,dA\right)\epsilon^2 d\epsilon\,(\sigma_n\,Y_2\,A)}{\sum_{b,c}\frac{s_{n-2\,b\,c}(\sigma_n\,b\,c)}{(\sigma_n\,b\,c)}\left(1-\epsilon\frac{(A\,b\,c)}{(\sigma_n\,b\,c)}\right)\sum_{b,c}\frac{s_{n-2\,b\,c}(Y_2\,b\,c)}{(\sigma_n\,b\,c)}\left(1-\epsilon\frac{(A\,b\,c)}{(\sigma_n\,b\,c)}\right)}$$
$$\times\left[\frac{(n-4\,n-3\,n-1)(n-3\,n-1\,1)(n-1\,1\,2)}{\epsilon(n-4\,n-3\,\sigma_n)(n-3\,\sigma_n\,n-1)(A\,n-1\,\sigma_n)(n-1\,\sigma_n\,1)(\sigma_n\,1\,2)}\right]^2. \quad (A.22)$$

By momentum conservation $\sum_{b,c\neq n} s_{n-2\,b\,c} + \sum_{c\neq n} s_{n-2\,n\,c} = 0$, therefore we can neglect $\sum_{b,c\neq n} s_{n-2\,b\,c}$, which is of $\mathcal{O}(\tau^2)$, in comparison with $\mathcal{O}(\tau)$ terms. We then obtain,

$$S_3^{(3)} = -\oint \frac{\left(\sigma_n\,d^2\sigma_n\right)(X_1\,Y_1\,\sigma_n)}{\sum_{b,c}\frac{s_{n\,b\,c}(X_1\,b\,c)}{(\sigma_n\,b\,c)}\sum_{b,c}\frac{s_{n\,b\,c}(Y_1\,b\,c)}{(\sigma_n\,b\,c)}}\oint\frac{\left(\sigma_n\,A\,dA\right)d\epsilon\,(\sigma_n\,Y_2\,A)}{\epsilon\sum_{b,c}\frac{s_{n-2\,b\,c}(A\,b\,c)}{(\sigma_n\,b\,c)}\sum_{b,c}\frac{s_{n-2\,b\,c}(Y_2\,b\,c)}{(\sigma_n\,b\,c)}}$$
$$\times\left[\frac{(n-4\,n-3\,n-1)(n-3\,n-1\,1)(n-1\,1\,2)}{(n-4\,n-3\,\sigma_n)(n-3\,\sigma_n\,n-1)(A\,n-1\,\sigma_n)(n-1\,\sigma_n\,1)(\sigma_n\,1\,2)}\right]^2. \quad (A.23)$$

Now consider $\sigma_n \to \sigma_{n-1}$ with the parametrization,

$$\sigma_n = \sigma_{n-1} + \eta B. \quad (A.24)$$

We will choose $X_1 = \sigma_{n-1}$ and also $Y_2 = \sigma_{n-1}$. With this choice we have,

$$
\begin{aligned}
S_3^{(3)} &= -\oint \frac{\eta^2 d\eta\,(n-1\,B\,dB)(n-1\,Y_1\,B)}{\displaystyle\sum_{b,c\neq n-1}\frac{s_{n\,b\,c}(n-1\,b\,c)}{(n-1\,b\,c)}\sum_c\frac{s_{n\,n-1\,c}(Y_1\,n-1\,c)}{\eta(B\,n-1\,c)}} \\
&\times \oint \frac{d\epsilon\,(n-1\,A\,dA)\,\eta\,(B\,n-1\,A)}{\epsilon\displaystyle\sum_c\frac{s_{n-2\,n-1\,c}(A\,n-1\,c)}{\eta(B\,n-1\,c)}\sum_{b,c\neq n-1}\frac{s_{n-2\,b\,c}(n-1\,b\,c)}{(n-1\,b\,c)}} \\
&\times \left[\frac{(n-4\,n-3\,n-1)(n-3\,n-1\,1)(n-1\,1\,2)}{\eta^3(n-4\,n-3\,n-1)(n-3\,B\,n-1)(A\,n-1\,B)(n-1\,B\,1)(n-1\,1\,2)}\right]^2 \\[2mm]
&= -\oint \frac{d\eta\,(B\,n-1\,dB)(Y_1\,n-1\,B)}{\eta\displaystyle\sum_{b,c\neq n-1}s_{n\,b\,c}\sum_c\frac{s_{n\,n-1\,c}(Y_1\,n-1\,c)}{(B\,n-1\,c)}}\oint \frac{d\epsilon\,(A\,n-1\,dA)(A\,n-1\,B)}{\epsilon\displaystyle\sum_c\frac{s_{n-2\,n-1\,c}(A\,n-1\,c)}{(B\,n-1\,c)}\sum_{b,c\neq n-1}s_{n-2\,b\,c}} \\
&\times \left[\frac{(n-3\,n-1\,1)}{(n-3\,n-1B)(A\,n-1\,B)(B\,n-1\,1)}\right]^2. \qquad\text{(A.25)}
\end{aligned}
$$

We can now take $\sigma_{n-1}$ as a spectator, and choose $Y_1 = \begin{pmatrix}0\\1\end{pmatrix}$, $A = \begin{pmatrix}1\\x_A\end{pmatrix}$, and $B = \begin{pmatrix}1\\x_B\end{pmatrix}$. Thus we have,

$$
\begin{aligned}
S_3^{(3)} &= -\frac{1}{\displaystyle\sum_{b,c\neq n-1}s_{n\,b\,c}\sum_{b,c\neq n-1}s_{n-2\,b\,c}}\oint \frac{(B\,n-1\,dB)(Y_1\,n-1\,B)}{\displaystyle\sum_c\frac{s_{n\,n-1\,c}(Y_1\,n-1\,c)}{(B\,n-1\,c)}} \\
&\times \oint \frac{(A\,n-1\,dA)(A\,n-1\,B)}{\displaystyle\sum_c\frac{s_{n-2\,n-1\,c}(A\,n-1\,c)}{(B\,n-1\,c)}}\left[\frac{(n-3\,n-1\,1)}{(n-3\,n-1\,B)(A\,n-1\,B)(B\,n-1\,1)}\right]^2 \\[2mm]
&= -\frac{1}{\displaystyle\sum_{b,c\neq n-1}s_{n\,b\,c}\sum_{b,c\neq n-1}s_{n-2\,b\,c}}\oint \frac{dx_B}{\displaystyle\sum_c\frac{s_{n\,n-1\,c}}{(x_c-x_B)}}\oint \frac{dx_A\,(x_B-x_A)}{\displaystyle\sum_c\frac{s_{n-2\,n-1\,c}(x_c-x_A)}{(x_c-x_B)}} \\
&\times \left[\frac{(x_1-x_{n-3})}{(x_B-x_{n-3})(x_B-x_A)(x_1-x_B)}\right]^2. \qquad\text{(A.26)}
\end{aligned}
$$

Taking $x_A = x_B + \epsilon$ and performing the integrations we get,

$$
S_3^{(3)} = \frac{-1}{\left(\displaystyle\sum_{b\neq n-1}s_{n-1\,n\,c}\right)\left(\displaystyle\sum_{c\neq n-1}s_{n-2\,n-1\,c}\right)^2}\left(\frac{1}{s_{n-3\,n-1\,n}}+\frac{1}{s_{n-1\,n\,1}}\right). \qquad\text{(A.27)}
$$

It can be checked, by explicit computations, that the other possibilities, namely, $(i)\,\sigma_n\to\sigma_1$, $(ii)\,\sigma_n\to\sigma_2$, $(iii)\,\sigma_n\to\sigma_{n-3}$, $(iv)\,\sigma_n\to\sigma_{n-4}$, will not contribute.

## A.2 Collinear singularities

This singularity lies at the codimension 2 boundary where $\sigma_{n-2}$ is at the intersection of two lines, one of them containing $\sigma_{n-4}$ and $\sigma_{n-3}$, and other line containing $\sigma_{n-1}$ and $\sigma_n$. Let $\xi$ be the point of intersection of these two lines then,

$$
(n-4\,n-3\,\xi)=0, \qquad (\xi\,n-1\,\sigma_n)=0. \qquad\text{(A.28)}
$$

We choose the parametrization as,

$$\sigma_{n-2} = \alpha\sigma_{n-1} + \beta\sigma_{n-3} + \xi. \tag{A.29}$$

The $\sigma_{n-2}$ integration becomes,

$$\oint \frac{(\xi\,n-1\,n-3)\,d\alpha\,d\beta}{\left[\frac{s_{n-2\,n-3\,n-4}(X_2\,n-3\,n-4)}{\alpha(n-1\,n-3\,n-4)} + \sum\limits_{b,c\neq(n-3,n-4)} \frac{s_{n-2\,b\,c}(X_2\,b\,c)}{(\xi\,b\,c)}\right]}$$
$$\times \frac{(X_2\,Y_2\,\alpha\sigma_{n-1} + \beta\sigma_{n-3} + \xi)}{\left[\frac{s_{n-2\,n-3\,n-4}(Y_2\,n-3\,n-4)}{\alpha(n-1\,n-3\,n-4)} + \sum\limits_{b,c\neq(n-3,n-4)} \frac{s_{n-2\,b\,c}(Y_2\,b\,c)}{(\xi\,b\,c)}\right]}$$
$$\times \left[\frac{1}{\underline{\alpha\beta}\,(n-3\,\xi\,n-1)(n-3\,n-1\,\sigma_n)}\right]^2. \tag{A.30}$$

We choose the reference vectors as $X_2 = \sigma_{n-1}$ and $Y_2 = \xi$ to obtain,

$$\frac{1}{s_{n-2\,n-3\,n-4} \sum\limits_{b,c\neq(n-3,n-4)} s_{n-2\,b\,c}} \left[\frac{1}{(n-3\,n-1\,\sigma_n)}\right]^2. \tag{A.31}$$

The $\sigma_n$ integration then takes the form,

$$S_{\text{col}}^{(3)} = \frac{1}{s_{n-2\,n-3\,n-4} \sum\limits_{b,c\neq(n-3,n-4)} s_{n-2\,b\,c}}$$
$$\times \oint \frac{(\sigma_n\,d^2\sigma_n)(X_1\,Y_1\,\sigma_n)}{\sum\limits_{b,c} \frac{s_{n\,b\,c}(X_1\,b\,c)}{(\sigma_n\,b\,c)} \sum\limits_{b,c} \frac{s_{n\,b\,c}(Y_1\,b\,c)}{(\sigma_n\,b\,c)}} \left[\frac{(n-3\,n-1\,1)(n-1\,1\,2)}{(n-3\,n-1\,\sigma_n)(n-1\,\sigma_n\,1)(\sigma_n\,1\,2)}\right]^2$$

$$= \left[\frac{1}{s_{n-2\,n-3\,n-4} \sum\limits_{b,c\neq(n-3,n-4)} s_{n-2\,b\,c}}\right]$$
$$\times \left[\frac{1}{s_{n-3\,n-1\,n}s_{n\,1\,2}} + \frac{1}{\sum\limits_{c\neq n-1} s_{n-1\,n\,c}}\left(\frac{1}{s_{n-3\,n-1\,n}} + \frac{1}{s_{n-1\,n\,1}}\right)\right.$$
$$\left. + \frac{1}{\sum\limits_{c\neq 1} s_{n\,1\,c}}\left(\frac{1}{s_{n\,1\,2}} + \frac{1}{s_{n-1\,n\,1}}\right)\right]. \tag{A.32}$$

Now the double soft factor for the non-degenerate configuration where $n$-th and $(n-2)$-th external states are going soft is given as,

$$S_{\text{non-deg}}^{(3)} = S_1^{(3)} + S_{2;1}^{(3)} + S_{2;2}^{(3)} + S_3^{(3)} + S_{\text{col}}^{(3)}. \tag{A.33}$$

We find that the above soft factor scales as $\mathcal{O}(\tau^{-4})$, which is subleading compared to the degenerate case.

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
