# Peer review of "Double Soft Theorem for Generalised Biadjoint Scalar Amplitudes"

_SciPost Physics, doi:SciPost Phys. 10, 036 (2021)_

## Round 2 · Referee Report · Anonymous (Referee 1) · 2020-12-6

Strengths

well organized, systematic

Weaknesses

emphasis on reporting computations vs interpreting them

Report

This paper presents a useful cataloging of double soft limits in an interesting theory: the generalized biadjoint scalar. The composition and computations are clear. However, the motivation and discussion could be strengthened. Under a strict reading of the SciPost acceptance criteria, it would seem that the closest requirement is option 3. "Open a new pathway in an existing or a new research direction, with clear potential for multipronged follow-up work." The introduction and discussion seem to present a fair case for the proposal of CEGM to meet this criterion, but less so for the results herein.

Requested changes

I think it would behoove the authors to augment their motivation / discussion in a manner that makes it clearer that publishing in SciPost is a good fit.

  • validity: top
  • significance: ok
  • originality: good
  • clarity: good
  • formatting: perfect
  • grammar: perfect

Author:  Md Abhishek  on 2020-12-12  [id 1075]

(in reply to Report 1 on 2020-12-06)
Category:
remark

We thank the referee for useful suggestions to strengthen our arguments regarding the significance of our results for the generalized bi-adjoint scalar theory. Since the theory involves massless fields soft theorems offer a window into the low energy physics. Developing this understanding may eventually help us find an appropriate field theory formulation for these theories. We would like to point out that, as of now, there is no efficient technique available to compute the CEGM amplitudes for arbitrary k and n. However the analytic results derived in this manuscript for the adjacent double soft limits provide a window to find the amplitudes for arbitrary k and n. Studying the soft factorisation helps us to understand the singularity structures of the amplitudes and give us the information about the boundaries of the moduli space of the underlying theory. Bootstrapping the soft amplitudes can, in principle, help us reconstruct the full amplitude and eventually provide a Lagrangian formulation for these theories for arbitrary k and n. Whether such an inversion is possible for any choice of (k>2, n) pair is an interesting open question. We intend to elaborate on explanations along these lines in the introduction and the discussion sections. We hope that this addresses the concern raised by the referee regarding the manuscript.

---

## Round 2 · Referee Report · Anonymous (Referee 2) · 2021-1-14

Strengths

Very well written and well organised . All the Computations are sufficiently detailed and carefully explained . Final result is very interesting as it may further enhance our understanding of this new class of theories based on Tropical Grassmannian .

Weaknesses

Motivation to study soft theorems in these theories not entirely clear .
From the introduction of the paper, one of the motivations seems to be that the soft structure may aid in probing a field theoretic formulation of these amplitudes.
However how do double soft theorems help in this regard is not entirely clear .
The conclusion section has a remark on how the presence of higher order poles indicate that these amplitudes could be relevant in loop diagrams of ordinary ($k = 2$) QFTs, but it will be nice if the authors expand the reason behind this intuition.

Report

The paper deserves to be published in scipost . Below I outline a number of suggestions that authors may take into account as I feel addressing them may help placing this work in a broader context . However these suggestions should be considered as optional and the paper may be published as is if the authors do not find them relevant.

(1) In 2019, Guevara et al studied single soft theorems for generalised bi-adjoint amplitudes and provided a geometric understanding of the soft factor. Although their analysis was done by using the positive Grassmannian formulation (something the authors do not use in this paper) .

However it will be nice if some of the results of this paper could be connected to that work. In particular, Guevara et al showed that for any $k$ the leading soft factor could be interpreted as the amplitude for ordinary bi-adjoint scalar field theory with $k+2$ particles . Even though this result was derived by understanding soft limits as embedding of certain facets into tropical Grassmannian, the final result (as stated above) can be understood without appealing to the Tropical Grassmannian picture.

Hence, it will be nice to understand if the double soft theorems authors have derived also have such an interpretation as being amplitudes of some theory. For next to adjacent external states, where the double soft factors seem to be product of single soft factors, such an interpretation may be possible and perhaps immediate.
And it will be extremely interesting if authors had any thoughts or analysis on interpreting the simultaneous double soft factors (when soft legs are adjacent) as amplitudes in their own right .
In other words, can these soft factors be understood as amplitudes of some bi-adjoint theory (either ordinary or generalised with $k^{\prime}$ being less then $k$) .

(2) To repeat my concerns written in the ``weakness" section, one way the paper could be strengthened is by motivating the question further. Given the insights one learnt from single soft theorems in these theories, what new could be learnt from various double and multi-soft theorems ?

---

## Round 3 · Author Response

We thank both the referees for their comments. In the introduction, as desired by the referees we have modified the third paragraph where we have elaborated on the motivation to study the double soft limit. Final paragraph of the discussion section has been modified where we point to some connections with one loop integrands derived by some of us. The cited article also addresses the double soft theorem from the perspective of Grassmannian cluster algebra which is a question raised by the second referee.

---

## Round 3 · List of Changes

1) Third paragraph of the introduction is new.

2) Final paragraph in the discussion section is modified.

3) Reference (69) is added.

You are currently on this page

Resubmission 2008.07271v3 on 31 January 2021

---

## Editorial Decision

published